# Dopamine release in human associative striatum during reversal learning

Filip Grill [1,2] ✉, Marc Guitart-Masip[3,4,5,6], Jarkko Johansson [1,2], Lars Stiernman[2,7], Jan Axelsson[2,8], Lars Nyberg [1,2,7] & Anna Rieckmann [1,2,7,9] ✉

The dopaminergic system is firmly implicated in reversal learning but human measurements of dopamine release as a correlate of reversal learning success are lacking. Dopamine release and hemodynamic brain activity in response to unexpected changes in action-outcome probabilities are here explored using simultaneous dynamic [11C]Raclopride PET-fMRI and computational modelling of behavior. When participants encounter reversed reward probabilities during a card guessing game, dopamine release is observed in associative striatum. Individual differences in absolute reward prediction error and sensitivity to errors are associated with peak dopamine receptor occupancy. The fMRI response to perseverance errors at the onset of a reversal spatially overlap with the site of dopamine release. Trial-by-trial fMRI correlates of absolute prediction errors show a response in striatum and association cortices, closely overlapping with the location of dopamine release, and separable from a valence signal in ventral striatum. The results converge to implicate striatal dopamine release in associative striatum as a central component of reversal learning, possibly signifying the need for increased cognitive control when new stimuli-responses should be learned.

Learning, unlearning, and relearning action-outcome associations are necessary to optimize gains and minimize losses in uncertain environments. To perform optimally, a balance must be struck between decision flexibility and rigidity; being too flexible can lead to maladaptive behavioral changes triggered by environmental noise while being too rigid can lead to stereotyped behaviors and missed opportunities. Probabilistic reversal learning paradigms are used to investigate decision flexibility under the reinforcement learning framework.

Using reinforcement learning, agents learn to perform actions based on predicted outcomes. Errors in the prediction are the canonical teaching signals in reinforcement learning and guide behavioral reversals[1,2]. Better than expected outcomes, positive reward prediction

errors (RPE), reinforce the actions that led to reward. By contrast, negative RPEs are important for learning because they signal a need to explore alternative options to achieve higher rewards in the future. Continuing with a previously rewarded choice after a reversal constitutes perseverance errors and are, by definition, associated with negative RPEs. Human neuroimaging studies have shown that the striatum is involved in processing RPEs[3,4] and there is considerable animal evidence that midbrain dopamine (DA) neurons fire in a pattern that is consistent with RPEs:[5] increased firing of DA neurons to outcomes that are more positive than expected and suppression of DA response to outcomes that are more negative than expected. However, it has been suggested that subpopulations of DA neurons operate

[1]Department of Diagnostics and Intervention, Diagnostic Radiology, Umeå University, Umeå, Sweden. [2]Umeå Center for Functional Brain Imaging, Umeå University, Umeå, Sweden. [3]Aging Research Center, Department of Neurobiology, Care Sciences and Society, Karolinska Institutet, Stockholm, Sweden. [4]Center for Psychiatry Research, Region Stockholm, Stockholm, Sweden. [5]Center for Cognitive and Computational Neuropsychiatry (CCNP), Karolinska Institutet, Stockholm, Sweden. [6]Max Planck UCL Centre for Computational Psychiatry and Ageing Research, University College London, London, UK. [7]Department of Medical and Translational Biology, Umeå University, Umeå, Sweden. [8]Department of Diagnostics and Intervention, Radiation Physics, Umeå University, Umeå, Sweden. [9]Institute for Psychology, University of the Bundeswehr Munich, Neubiberg, Germany. ✉e-mail: filip.grill@umu.se; anna.rieckmann@unibw.de

differently and instead fire more generally in response to unexpected, salient, events irrespective of stimuli value (e.g. ref. [6–10],). Bromberg-Martin et al. proposed that neurons that release DA to both reward and punishment support dorsal striatal brain systems for attention, working memory, and general motivation whereas DA neurons that comply with canonical RPEs represent subjective stimuli value and support ventral striatal brain systems for value learning[11]. In line, Ishino et al. found that DA neurons in the midbrain of rodents that project to the dorsal parts of the nucleus accumbens (near the border of caudate) signal learning from lack of expected rewards[9].

In humans, systemic DAergic drugs have been shown to modulate RPEs[12], increase sensitivity to RPEs[13–15], and influence reversal learning performance[15,16]. Together, human and animal research converge to implicate the DAergic system in reversal learning from RPEs but human in vivo imaging evidence that spatially and temporally associates striatal DA release with the encounter of RPEs and evidence for individual differences in DA release as a correlate of reversal learning success is lacking.

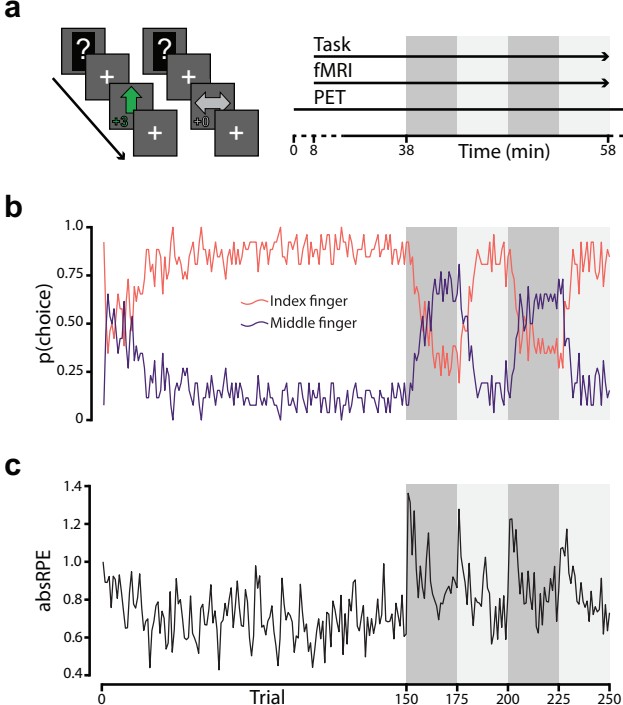

**Fig. 1 | Paradigm structure, choice probability, and model based absRPE magnitude. a** Participants performed a two forced-choice reversal learning task. Each trial was initiated with a question mark prompting the participants to make a choice of either answering with their index or middle finger. If their choice was correct, they were rewarded with 3 SEK, if incorrect they received no reward. For the first 150 trials, answering with their index finger was rewarded 80% of the time while answering with their middle finger was rewarded 20% of the time, constituting a stable task period. After 150 trials, reward contingencies were reversed (dark gray area), after 25 trials reward contingencies reversed back (light gray area), the reward contingencies then kept reversing every 25 trials until the task ended, constituting a volatile task period. [11C]Raclopride PET and fMRI BOLD imaging were simultaneously collected during the whole task. **b** There was an initial response bias of choosing the index finger which over a couple of trials resolved to around 50% probability (possibly reflecting initial exploration). The probability of choosing the index finger then steadily increased to reflect the reward contingencies during the stable task period. The choice probability then fluctuated according to the reversal manipulations. **c** Modelling the behavioral data according to a reinforcement learning model showed marked increases in absRPE magnitude on the group level coinciding with each reversal. Source data are provided as a Source Data file. fMRI - functional Magnetic Resonance Imaging, PET – Positron Emission Tomography, absRPE – Absolute Reward Prediction Error.

The primary method to measure DAergic function in humans is positron emission tomography (PET). [11C]Raclopride, an antagonist for the DA D2/D3 receptors[17], is a well validated and commonly used radioligand for imaging of striatal DA release. The binding affinity of [11C]Raclopride is lower than that of endogenous DA, enabling receptor occupancy competition[18]. Through this competition principle, DA release at the scale of 20–30 min can be estimated for pharmacological challenges[19,20], and by comparing two cognitive states[21–27]. Hybrid PET and MR imaging enables simultaneous acquisition of [11C]Raclopride PET and blood-oxygen-level-dependent (BOLD) fMRI, which provides synergistic information regarding neurochemical signaling and hemodynamic responses at different timescales. Whether the magnitude of DA release is proportional to fMRI signal magnitude is an open question because fMRI is an indirect measure not only reflective of neurotransmitter action but also many other signaling cascades.

In this study, we utilize a two-forced choice learning paradigm that contained unexpected rule reversals at certain periods. A 30-min stable period, where rewards were easy to obtain as one choice was rewarded 80% of the time, was followed by a 20-min volatile period, where the most rewarded choice changed every 5 min. If participants correctly tracked the most rewarded choice, participants were likely to encounter predictive errors after the transition from the stable to volatile period. Consistent with the work that has linked DA release in dorsolateral parts of the midbrain DA system to both rewarding and punishing stimuli, we observe DA release from dynamic [11C]Raclopride PET in dorsal, associative striatum[28] at the timepoint of transition from stable to volatile period. Absolute RPEs (absRPE, a signal that has been previously associated with salience and surprise[4]) were estimated from computational modeling of task performance and used to estimate whether the magnitude of DA release in associative striatum correlates with the magnitude of absRPE and general sensitivity to RPEs. We further predicted that the site of DA release would be congruent with trial-based fMRI responses to perseverance errors and absRPE, that these also include a cortical cognitive control system, and that these are separable from event-related fMRI responses to valence in ventral striatum.

## Results
### Paradigm and behavior
To enable experimental control in a paradigm that fits both PET and fMRI, we used a two-forced choice reversal-learning task with a long stable period followed by a volatile period (Fig. 1a). Throughout the task, participants were asked to guess if a hidden number was above (index finger) or below 5 (middle finger). If correct, they were rewarded, and if incorrect they received nothing. Unbeknownst to the participants, the task was rigged such that answering above 5 was rewarded 80% of the time during the stable period. After transitioning to the volatile period, answering below 5 was instead rewarded 80% of the time. In the volatile period, the reward contingencies reversed every 5 min. The PET data was modeled to estimate DA release at the first reversal, i.e., the transition between stable and volatile phase, as dopamine is assumed to peak at the start of the volatile period. The fMRI analysis modeled individual events during the entire task. Cue, response and outcome response within each event were not separable.

The average choice probability over the group confirmed that the reversal manipulation was effective, with gradually decreasing probability of the previously most rewarded choice after reversal transitions (Fig. 1b). The mean ($M = 5.84$ trials) and the standard deviation ($SD = 6.79$ trials) of the perseverance error (i.e., the number of trials a participant kept to the previously most rewarded choice) for the first reversal showed that there were large individual differences in how quickly individuals were able to re-learn the action-outcome association.

To estimate RPE magnitude and associated parameters related to task performance, we fitted a series of reinforcement learning models

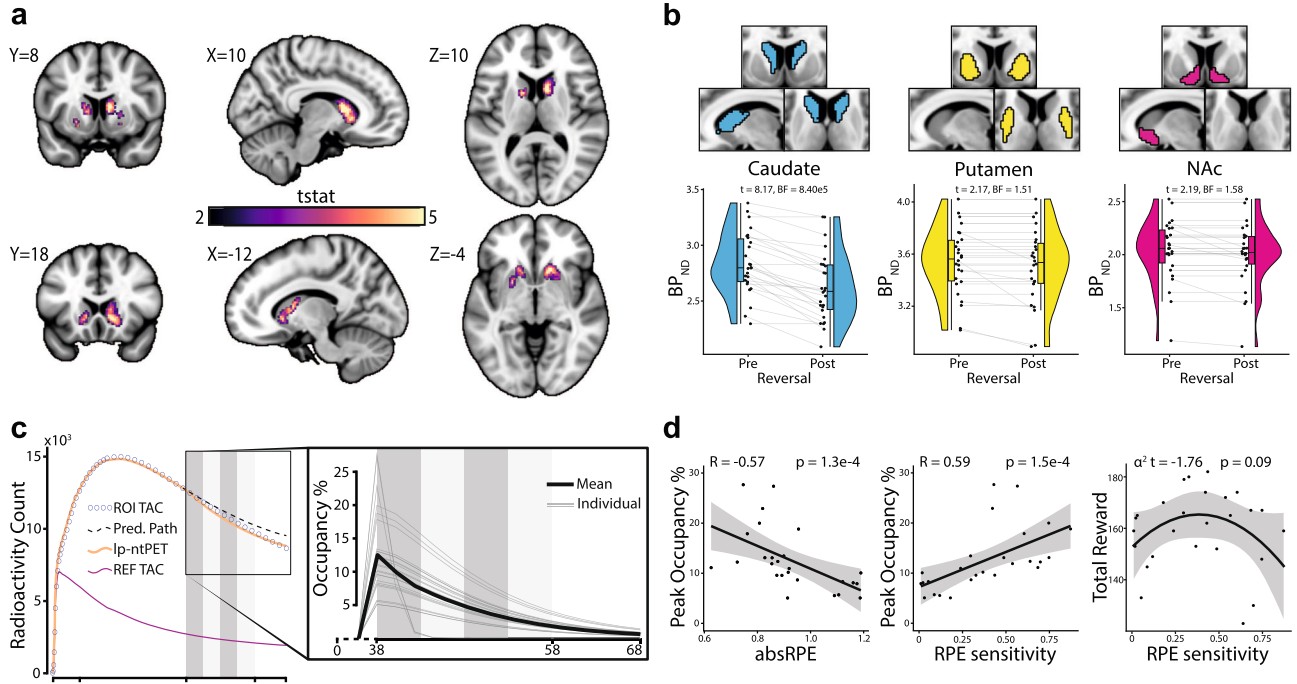

**Fig. 2 | Dopamine release cluster, dopamine receptor occupancy, and behavioral associations. a** Voxelwise group level result of the lp-ntPET analysis ($N = 26$ participants; peak MNI coordinate = xyz[10,10,10], TFCE corrected one-tailed one sample $t$-test: peak t(25) = 6.30, $p < 0.002$). The bilateral clusters represent voxels with significant DA release (color bar represents t-statistics; MNI-coordinates). **b** As a control analysis the lp-ntPET model was fitted on a priori defined anatomical ROIs in the striatum ($N = 26$ participants). Congruent with the voxelwise analysis, using Bayes Factor two-tailed paired samples $t$-test a decrease in $BP_{ND}$ was observed in the caudate during the transition from stable (pre) to volatile (post) task period while the putamen and NAc $BP_{ND}$ decrease was not observed for the majority of participants. Boxplots show the median and 25th and 75th percentiles, with the whiskers extending max. 1.5 * interquartile range. **c** For illustration purposes, the mean TAC from the significant voxels, lp-ntPET fit, predicted path (i.e. the path of the TAC if no DA was released), and reference TAC (cerebellum) over the whole group is depicted. The zoomed in box shows the fitted occupancy functions at the critical reversal period for each individual as well as the mean over over the group. DA receptor occupancy peaked at the first reversal for all participants. It should be noted that at

this point in the experiment the PET frames had a 2 min duration as well as some temporal smoothing, which is why it appears as if DA receptor occupancy increases before the reversal. Dark grey and light grey indicate reversal onsets. **d** Peak DA receptor occupancy was correlated with mean RPE magnitude over a 25 trial windows after the first reversal ($N = 26$ participants; Pearson correlation: r(24) = −0.57, $p = 0.003$) and with RPE sensitivity ($N = 26$ participants; Pearson correlation: r(24) = 0.59, $p = 0.0015$; note that RPE sensitivity was estimated on all trials). An inverted-U association was observed between RPE sensitivity and total reward ($N = 26$ participants; linear regression: t(23) = −1.76, $p = 0.09$) though nonsignificant for this behavioral model the effect was consistent across the model space (shaded areas represent 95% confidence interval). Source data are provided as a Source Data file. $BP_{ND}$ Binding Potential, BF Bayes Factor, NAc Nucleus Accumbens, ROI Region Of Interest, Pred. Predicted, lp-ntPET Linear Parametric Neuro-Transmitter Positron Emission Tomography, REF Reference region, TAC Time Activity Curve, absRPE absolute Reward Prediction Error, RPE Reward Prediction Error.

to the observed behavioral data. The best performing computational behavioral model was a simple model that fits two free parameters to the data: β (inverse temperature of choice probability; Supplementary Equation 1), and α (RPE sensitivity, often referred to as learning rate; Supplementary Equation 2). Large individual differences were observed in the fitted parameters (α: $M = 0.38$, SD = 0.26; β: $M = 3.20$, SD = 4.64). Supplementary information on model selection, recovery analyses, and model behavior can be found in Supplementary Table 1 and 2, and Supplementary Fig. 3. In order to disentangle processing of unexpected events (reward & no reward) from valence (reward vs no reward) in the second fMRI event-related design, we chose absRPE over signed RPE as a metric of interest throughout. However, by design, absRPEs are largely reflective of negative RPE for the period immediately after the reversal of a learned association.

The mean absRPE magnitude across participants showed an expected increase coinciding with the reversal transitions (Fig. 1c). Linear regressions showed that the mean absRPE magnitude over 25 trials after the first reversal was significantly associated with the number of perseverance errors (F(1,24) = 6.78, $p = 0.0156$, R2 = 0.19) such that for every 0.1 increase in mean absRPE magnitude after reversal, perseverance errors were prolonged with 2.06 trials. By definition of how the computational model was set up, absRPE

magnitude is related to RPE sensitivity (F(1,24) = 47.20, $p = 4.18e-7$, R2 = 0.65) such that for every 0.1 increase in mean absRPE magnitude, RPE sensitivity decreased by 0.14. Thus, individuals with low absRPE magnitude over 25 trials had less perseverance errors, indicative of faster re-learning, and were more sensitive to absRPEs.

## Striatal dopamine release during reversal learning

Voxelwise PET time activity curves (TAC) in striatum were modeled with a set of time-varying basis functions to identify sites of perturbed radiotracer binding[23–26,29–31] in relation to the transition from stable to volatile task period. On the group level, voxelwise analyses showed that a bilateral cluster, mostly located in the caudate, exhibited the strongest tracer displacement as inferred from binding potential ($BP_{ND}$) differences (peak MNI coordinate = xyz[10,10,10], peak t(25) = 6.30, k-voxels = 828, $p = 0.0002$, threshold-free cluster enhancement (TFCE) corrected; Fig. 2a). A post-hoc anatomically defined region of interest (ROI) analysis using Bayes Factor paired samples $t$-test showed that consistent displacement was only observed in the caudate ($t = 8.17$, BF = 8.40e5); no changes were detected in the putamen ($t = 2.17$, BF = 1.51) nucleus accumbens ($t = 2.19$, BF = 1.58) (Fig. 2b). Next, to estimate individual differences in DA receptor occupancy by endogenous DA (i.e., DA release inferred by differences

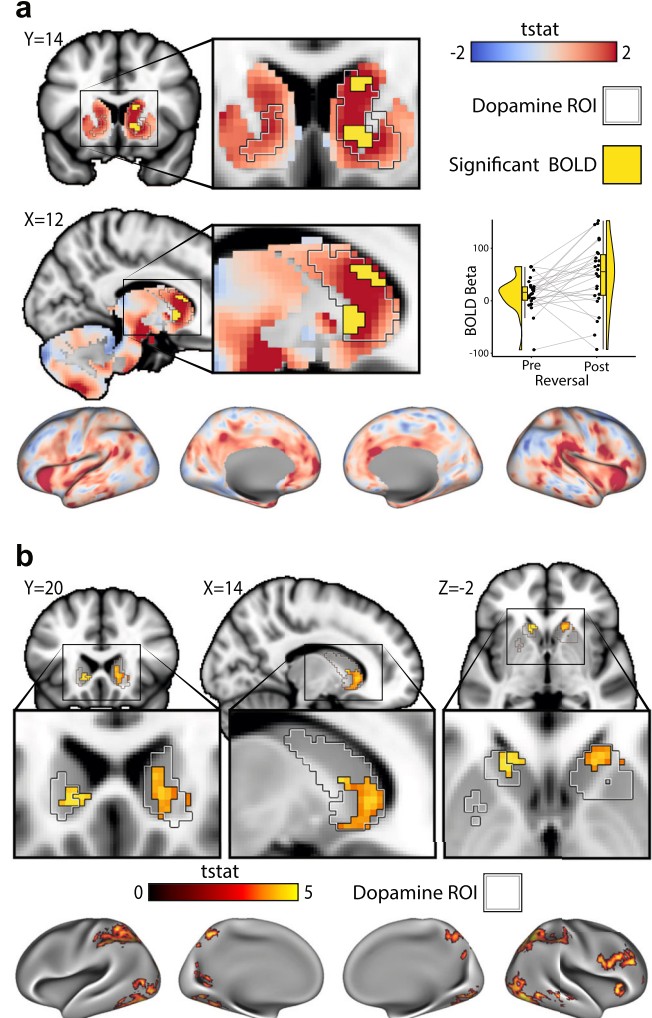

**Fig. 3 | BOLD responses from fMRI. a** Significant voxels of the perseverance error contrast ($N = 26$; peak MNI-coordinate = xyz[12,16,14], small volume TFCE corrected one-tailed one-sample *t*-test: peak t(25) = 3.11, $p = 0.0316$) in yellow inside the significant DA release ROI (black and white outline; MNI-coordinates). The voxels show a significant difference in BOLD response between reward trials before the reversals and perseverance error trials after the reversal, i.e., participants are making the same choice but the outcome is different. Whole brain t-statistics map is displayed in blue to red. Boxplots show the median and 25th and 75th percentiles, with the whiskers extending max. 1.5 * interquartile range and are there to visualize the BOLD response of the contrast. **b** Whole brain BOLD fMRI response that covary with trial by trial RPEs as estimated from the computational cognitive model. BOLD response was observed in the striatum ($N = 26$ participants; peak MNI-coordinate = xyz[−12,20,−2], TFCE corrected one-tailed one-sample *t*-test: t(25) = 4.37, $p = 0.0432$), mostly confined to the voxels where DA release was located. In the cortex, BOLD response of absRPE were observed in the right anterior insula ($N = 26$ participants; peak MNI-coordinate = xyz[36,22,2], t(25) = 4.75, $p = 0.0346$, TFCE corrected), right dorsolateral prefrontal cortex (DLPFC; $N = 26$ participants; peak MNI-coordinate = xyz[48,34,22], t(25) = 5.43, $p = 0.0212$ TFCE corrected), bilateral parietal cortex ($N = 26$ participants; peak MNI-coordinate = xyz[−44,−42,46], t(25) = 6.08, $p = 0.0082$ TFCE corrected), and the occipital cortex ($N = 26$ participants; peak MNI-coordinate = xyz[−28,−88,14], t(25) = 4.98, $p = 0.031$, TFCE corrected) (MNI-coordinates). Source data are provided as a Source Data file. ROI Region Of Interest, BOLD Blood Oxygenation Level Dependent.

in pre-reversal $BP_{ND}$ and post-reversal $BP_{ND}$), the group-level displacement cluster was used as a ROI from which the average TAC was extracted for each individual. DA release was observed for all participants and peaked within 2 min during the transition from stable to volatile period (peak DA occupancy %: $M = 12.56$, SD = 6.32; Fig. 2c).

To confirm the behavioral relevance of striatal DA release, the peak individual DA receptor occupancy was correlated with the key behavioral parameters from the cognitive model. A significant negative association ($r = −0.56$, $p = 0.003$) was found between peak DA occupancy and absRPE magnitude over 25 trials at the corresponding reversal (Fig. 2d). Thus, individuals with low absRPE magnitude released more DA, indicating that the amount of DA release was related to how fast individuals reversed their choice. RPE sensitivity was significantly associated with peak DA occupancy ($r = 0.59$, $p = 0.0015$; Fig. 2c). The relationship between RPE sensitivity and task performance (total reward) was not linear ($r = −0.01$, $p = 0.95$). Instead, Fig. 2d suggests an inverted-U association between RPE sensitivity ($α^2$, $p = 0.09$) and task performance. These associations were independent of the computational model selection (Supplementary Fig. 1).

Importantly, neither pre-reversal $BP_{ND}$ nor post-reversal $BP_{ND}$ correlated with the behavioral measures, signifying the importance of measuring the DAergic system while it is active (Supplementary Fig. 2a). The association between occupancy and absRPE magnitude was robust to the choice of window size (Supplementary Fig. 2b).

## BOLD responses

Due to the temporal resolution of PET, the identification of DA release to errors was confined to the reversal period, where errors are maximized across participants. To capture concurrent BOLD activation at the first reversal, a primary BOLD contrast of interest was defined to compare rewarded correct responses on the 25 trials before the first reversal with perseverance errors over 25 trials after the first reversal (Supplementary Fig. 7a depicts regressors and contrast). In order to focus on voxels that overlap with the location of DA release, the DA release cluster was used as a ROI. Significant BOLD response differences were observed in this cluster in the right caudate (peak MNI-coordinate = xyz[12,16,14], t(25) = 3.11, k-voxels = 14, $p = 0.0404$, small volume TFCE corrected; Fig. 3a). A second analysis identified hemodynamic responses from fMRI to trial-level absRPEs encountered over the entire experiment and in whole brain. absRPEs were coded on the single-trial level according to the computational model and orthogonalized with respect to valence of each trial (Supplementary Fig. 7b depicts regressors and contrast). We reasoned that overlap of this second model with DA release supports a conclusion that links DA release to lack of rewards at reversal with the surprise component of RPEs more generally rather than to a change in valence. Moreover, because this second model was fit trial-by-trial, temporally removed from the PET model, overlap between DA release and activation in this second model would speak against a link between DA release and a slow contextual shift in volatility. BOLD response of trial-level absRPE were observed in the striatum (peak MNI-coordinate = xyz[−12,20,−2], t(25) = 4.37, $p = 0.0432$, TFCE corrected), mostly confined to the voxels where DA release was observed (Fig. 3b). In the cortex, BOLD response of absRPE were observed in the right anterior insula (peak MNI-coordinate = xyz[36,22,2], t(25) = 4.75, $p = 0.0346$, TFCE corrected), right dorsolateral prefrontal cortex (DLPFC; peak MNI-coordinate = xyz[48,34,22], t(25) = 5.43, $p = 0.0212$ (TFCE corrected)), bilateral parietal cortex (peak MNI-coordinate = xyz[−44,−42,46], t(25) = 6.08, $p = 0.0082$ TFCE corrected), and the occipital cortex (peak MNI-coordinate = xyz[−28,−88,14], t(25) = 4.98, $p = 0.031$, TFCE corrected; Fig. 3b). In this absRPE analysis, reward and no reward were added as nuisance regressors to control for stimulus valence. For completion, Supplementary Fig. 7c reports BOLD responses to the valence contrast (reward > no reward), identifying the canonical reward response in ventral striatum. DA release and its corresponding fMRI signals to absRPE, i.e., surprise more generally, are situated immediately adjacent but superior to the reward response. Correlations between individual differences in BOLD signal and DA occupancy were not significant (Supplementary Fig. 8).

## Discussion

In this study, we used simultaneously acquired dynamic [11C]Raclopride PET-fMRI and computational modeling to provide evidence for striatal DA release to lack of expected reward as a core component of reversal learning in humans. We detected significant DA release in associative striatum, congruent with a reward reversal and consistent with recent animal work that has established DAergic responses as a teaching signal of learning. BOLD activations to errors after reversal were identified in the location of DA release in associative striatum. An additional analysis of fMRI data to event-related absRPE throughout the task also shows that unexpected events per se, independent of value sign and not necessarily linked to a reversal period, are spatially consistent with the site of DA release and separable from a canonical reward response in ventral striatum.

The reversal learning task in this study was designed to bias participants' responses towards one choice over a long stable task environment after which the reward contingencies reversed and a volatile task environment begun. Behaviorally, the contextual shift produces perseverance errors during the transition. We found large interindividual variability in the number of perseverance errors during the first reversal, which is evidence for differences in how well established the response bias was. Low absRPE magnitude at reversal was associated with high RPE sensitivity (often referred to as learning rate in reinforcement learning literature) and both measures shared a high correlation with DA release. This indicates that high DA release supports a quick adaptation of behavior after an accumulation of errors, which is of advantage in a volatile environment. Conversely, low RPE sensitivity and little DA release at the reversal are related to a rigid response bias that is associated with slow adaptation of behavior in volatile environments. Of note, more behavioral adaptability is not always advantageous as evidenced by an inverted U shape association between RPE sensitivity and total reward obtained throughout the experiment. For example, flexible adaptions of behavior may be advantageous in volatile environments but detrimental in stable ones. In sum, through quantitative measurements of neurotransmitter release, we show that larger amounts of DA release are highly proportional to a faster reversal of behavior when learned associations change. Our work shows that it is the reactivity of the human DA system (i.e., the release in response to a specific change in error probability) and not level of DA receptor density that underlies flexibility in reversal learning. We also establish that neither rigid nor overly flexible learning is best for task success, suggesting a medium amount of DA release as optimal.

From our study it remains unclear what a medium amount of DA release generally constitutes and whether the magnitude is dependent on experimental design. Comparisons across studies suggest that our mean change in $BP_{ND}$ of 12.65 % is higher than the 5–10% estimates from PET studies of striatum looking at rewarding stimuli (e.g[21,32].), though on par with a study on conditioning[22]. How these estimates are comparable and what constitutes a normal amount of DA release need to be determined by studies that combine pharmacological control and task situations in the same design. Of note, a study by Boileau et al. showed comparable amounts of DA release either in response to amphetamine or to a (conditioned) placebo[20], suggesting that a relatively strong DA release in the context of our learning study appears plausible and is consistent with a general role for DA in learning about rewards rather than reward per se.

In contrast to prior work with pharmacological manipulations, genetics, or recordings in animals, this human in vivo imaging work allowed us to spatially locate DA release during reversal learning within striatum. Our study shows that increasing negative feedback during reversal learning (i.e., a lack of reward) selectively elicits DA release in associative striatum with corresponding neural activity in cortical cognitive control regions[33]. Animal work has predominantly linked reward, signed RPE and DA to the ventral striatum and mesolimbic system during positive RPEs. Our work instead suggests a model of human reversal learning in which DAergic responses to unexpected events activate the human mesocortical DA system, separable from reward-coding neuronal populations in ventral striatum, and thus more in line with new emerging animal work that has proposed DA release to errors[9]. In line with our observation of low absRPE magnitude after reversal being coupled with high DA release, it is possible that the observed DA release reflects a higher order cognitive-control process that is influenced by encountering unexpected events rather than absRPEs per se. The theory of opportunity cost models asserts that diminishing cognitive control occurs when the average reward rates are high[34]. Conversely, decreasing the average reward rate, as in the transition between stable and volatile period utilized in this study, should increase cognitive control[35], which has been linked to DA functions[36]. Since participants were not instructed that the task contained reversals, it is possible that the observed DA release reflects updating of an internal task model to include reversal monitoring. This view is congruent with our exploratory analyses finding no evidence for DA release of a similar magnitude and reliability during the volatile period (Supplementary Fig. 5). If absRPEs drive higher order cognitive processes in associative striatum, the two constructs would be highly correlated obstructing a clear disentanglement of the two.

Further support for a tight correspondence between absRPE and cognitive control was provided by simultaneously acquired fMRI data. The BOLD response to perseverance errors during the first reversal manipulation showed a spatial overlap with the striatal DA release cluster fitted at the same timepoint in the design. Extending the analysis beyond the reversal period with a trial-by-trial absRPE regressor over the whole task revealed that hemodynamic absRPE responses are also spatially congruent with the DA release cluster and extended to right anterior insula, bilateral parietal cortex, DLPFC. The insula, and parietal cortex have been described as part of the brain's attention network while the parietal cortex and DLPFC as part of a frontal control network[33,37]. Both network configurations have been shown to include parts of the caudate[38] in an area described to possibly mediate attentional control through the convergence of prefrontal and parietal cortical connections[39]. Additionally, the caudate has direct connections with the DLPFC[40] and is thought to be modulated by DA[41]. The observed DA release in the associative striatum might thus act to engage these networks when transitioning from a stable to volatile environment to facilitate cognitive control and error detection. Together, the results suggest that striatal DA release is a central component of reversal learning that might signal the need for cognitive control as environmental reward contingencies change.

What remains unclear from this human PET modeling work is whether and how DA PET signals and the BOLD signal are mechanistically linked, both where they overlap and where they do not overlap. Here, we explored the possibility of an association between BOLD and DA occupancy but found no significant correlations (Supplementary Fig. 8). A dose response relationship between raclopride and cerebral blood volume has previously been observed in non-human primates, providing ample evidence for neurovascular coupling of D2/D3 receptors[30]. However, using a ligand specific to D2/D3 receptors in humans with a cognitive task instead of selective pharmacological manipulations, we cannot conclusively speak to the influence of DA release on the BOLD signal since this is dependent on what type of receptor (D1-like or D2-like) DA binds to; in the current study it is likely that DA binds to a mix of receptor types. Moreover, also non-DAergic signals are captured in the BOLD signal. Thus, BOLD activations that are not accompanied by DA release might indicate additional neurotransmitter processes within a cortico-striatal network. Ideally, future hybrid PET-MR studies would implement multi-tracer studies or combine cognitive with pharmacological challenges in order to capitalize on the unique possibility of hybrid PET-fMRI to understand network-wide brain dynamic in terms of underlying neurotransmitter action.

There are also some limitations to the current study. Recent work has pointed out a number of potential methodological problems for single-scan designs in PET[42]. We found no confounding effects of head motion and adding control functions in the PET model (that fit DA release incongruent with the reversal) did not alter the spatial statistics. Further, the fact that the site of DA release is clearly localized to the associative striatum, overlaps with fMRI activity, and correlated with performance on the task, makes it unlikely that the conclusions we draw are driven by methodological confounds. However, a trade-off induced by our controlled experimental design is that we have no task-free resting state PET data to quantify model bias in our set-up.

Previous research on reversal learning has emphasized the modulation of RPE sensitivity when transitioning from stable to volatile environments[14,43]. In an attempt to address such modulation we fit a previously established model that allowed RPE sensitivity to be modulated trial by trial depending on choice confidence[44]. However, this model did not outperform a simple reinforcement learning model which estimates a single RPE sensitivity using all trials suggesting that increased model complexity was not justified by the data from this paradigm. Finally, limitations of the PET/MR gradient system necessitated a long TR in our experiment; future studies should consider sequence optimization in order to better jitter between cue and outcome for each trial and separate these events.

In conclusion, our study provides in vivo human multi-modal imaging data of striatal DA functions during reversal learning. Dynamic DA PET and fMRI data align to pinpoint the associative striatum as a site of DA release when unpredicted events are encountered. Critically, the amount of DA release during the transition from a stable to volatile environment was associated with better reversal learning but could lead to excessive decision flexibility in environmental contexts where decision flexibility is inappropriate. Taken together, our work suggests a model of human reversal learning in which DAergic responses to absRPEs activate the human mesocortical DA system, separable from reward-coding neuronal populations in ventral striatum.

## Methods

### Participants

Thirty participants were recruited through ads posted at the campus of Umeå University. Exclusion criteria consisted of current or past diagnosis of neurological or psychiatric illness, claustrophobia, history of head trauma, alcohol or drug dependence, and use of psychopharmaceuticals, drugs, or stimulants other than caffeine or nicotine for the past 6 months. Individuals with MRI-incompatible implants or objects were excluded for MRI safety reasons. Individuals that had previously undergone PET scanning for research purposes as well as pregnant or breast-feeding individuals were excluded for radiation safety reasons. Four participants were excluded from the study due to technical reasons relating to the timing of the PET/fMRI acquisition. The final sample thus consisted of twenty-six participants (13 female; mean age = 25.73; SD = 4.57; range = 20–36). All participants provided informed consent. Participants were compensated with 1000 SEK, up to an additional 600 SEK based on accumulated rewards during the experiment. This study was approved by the Regional Ethics Committee at Umeå University (2015/239-31).

### Study protocol

The participants were greeted at the Nuclear Medicine department at Umeå University Hospital and briefed regarding PET and MRI safety. Participants signed informed consent and were then trained on a short task resembling the in-scanner behavioral task but with random outcomes. Participants were then positioned in the PET/MR scanner and injected with [11C]Raclopride at the start of the PET acquisition. T1-weighted structural images were first acquired (Acquisition parameters: [FOV: 25 × 20 cm$^2$, matrix: 256 × 256, Slice Thickness: 1 mm, Slices: 180, TE: 3.1 ms, TR: 7,200 ms, Flip Angle: 12, Bandwidth:

244.1 Hz/Pixel], total time 7.36 min) while participants observed a fixation cross. Eight minutes after the PET scan started, the fMRI and behavioral task began. Behavioral and fMRI data were collected for 50 min. Finally, a B0 field map was collected. The total scan time was 68 min.

### Behavioral task

Participants performed a two-forced-choice reversal learning task. The task was created and displayed using PsychoPy2 1.85.2. Participants were shown a black card with a white question mark and instructed that behind the card was a number between 1 and 9. If they believed the number was above 5, they were instructed to respond with their index finger, instead if they believed the number was below 5, they should respond with their middle finger. The response window was 2 seconds, followed by a fixation cross of two seconds and the presentation of the outcome for 2 s. If they guessed correctly, a green arrow pointing upwards was displayed along with a text showing the amount won (+3). If they guessed incorrectly, a gray double headed arrow pointing to the right and left was displayed together with a text indicating that nothing was won (+0). The total accumulated rewards were also displayed at each outcome screen. An inter trial interval (ITI) consisting of a white fixation cross then followed with varying duration. The ITI durations were between 1–13 s and distributed in a pseudorandomized way tailored so that 25 trials took 5 min to complete. Each participant completed 250 trials of the task. Unbeknownst to the participants, there was no pre-specified number hidden behind the question mark. Instead, there was a probabilistic reward contingency such that for the first 150 trials there was an 80% chance of getting a reward if they responded with their index finger and a 20% chance of reward if they responded with their middle finger. Without a cue, these contingencies reversed after 150 trials and were then reversed every 25 trials creating a stable (first 150 trials) and volatile (last 100 trials) task environment. Note that the participants were unaware that any reward contingency reversals would occur. Participants were instructed that they would get the total accumulated rewards as extra payment (maximum 600 SEK).

### Computational model selection

Five models from prior work were selected as candidate computation models of behavior, with the main goal to extract trial-wise RPE and RPE sensitivity estimates. See supplementary materials for the full model space, model comparison (supplementary Table 1), parameter recovery (supplementary Table 2), model performance (supplementary Fig. 3a, b), and model recovery (supplementary Fig. 3c). Model fitting was performed using rSTAN (2.26.1). Signed RPEs obtained from the winning model were converted to unsigned, absolute RPE (absRPE). absRPE magnitude 25 trials after the transition from stable to volatile period (i.e. the first reversal) was averaged to quantify the magnitude of "unexpectedness" after reversal and a linear regression was performed between this average and number of perseverance errors during the same period. A linear regression was also performed between the average absRPE magnitude and individual estimations of RPE sensitivity. The average absRPE magnitude was correlated with peak DA receptor occupancy. Individual estimations of RPE sensitivity were also correlated with peak DA receptor occupancy, and a linear regression including a quadratic term for RPE sensitivity with total reward (proxy for task performance) as dependent variable was performed. These correlations were also performed on parameter estimates from the other candidate models as a control analysis (Supplementary Fig. 1).

### PET

All participants were injected with a bolus (250 MBq) of [11C]Raclopride following the local standard protocols for [11C]Raclopride PET studies[21,23,45]. A 68 min (6 × 10 s, 6 × 20 s, 6 × 40 s, 9 × 60 s, 26 × 120 s) dynamic time-of-flight acquisition and an MR-based attenuation

correction were collected. By employing a resolution recovery OSEM algorithm (3 iterations, 28 subsets, 3.0 mm post filter) with decay, randoms, scatter, and attenuation corrections, the data was reconstructed to a voxel size of $1.56 \times 1.56 \times 2.78$ mm$^3$. Each frame was motion corrected using FSL's (5.0) MCFLIRT[46] to the 25$^{th}$ frame with mutual information as cost function. A HYPR filter was applied to the data[47] and the data were temporally smoothed using a three-frame gaussian kernel ([0.25 0.50 0.25]). Each individual's T1w image was parcellated using FreeSurfer (6.0)[48], the parcellations were registered to the mean PET image and used to extract ROI TACs.

To identify areas of DA release, linear parametric neurotransmitter PET (lp-ntPET) was used to estimate voxelwise dynamic BP$_{ND}$ from voxelwise TACs[23,29–31] (model fitting was performed in MATLAB R2017b). A grey matter ROI from the cerebellum was used as a reference region and a multilinear reference tissue model with fixed k2' (estimated from the whole striatum) was conducted. In the main results, 5 different gamma basis functions were fitted, hypothesis-driven to be consistent with the transition of the first reversal. The best fitting function was then interpreted as accounting for [11C]Raclopride displacement occurring at the transition between periods. The basis-function approach permits inter-individual and inter-regional differences in [11C]Raclopride displacement, adaptive to the unknown shape of dynamic DA release.

Each voxels best solution resulted in individual parameter estimation maps representing [11C]Raclopride displacement due to DA release. The resulting parameter estimation maps were masked using a voxelwise F-statistics > 9.55 (Supplementary Equation 7) to control for false positive voxels on the individual level. The parameter estimation maps were normalized to MNI152 space and used in a second level analysis using FSL's randomise (5000 permutations; TFCE corrected) which estimated the group mean spatial distribution of [11C]Raclopride displacement using a one-tailed one-sample t-test. The output provided a statistical map of coherent spatial locations over the group of [11C]Raclopride displacement, interpreted as DA release, during the critical transition from stable to volatile period of the task.

In a next step, the significant [11C]Raclopride displacement cluster was used as an ROI to extract individual TACs for which the lp-ntPET model was fitted again to yield dynamic BP$_{ND}$ estimate curves which were transformed to DA receptor occupancy:

$$Occupancy(\%) = \frac{pre\,BP_{ND} - post\,BP_{ND}}{pre\,BP_{ND}} \times 100 \qquad (1)$$

This two-step approach is done to reduce the noise inherent in the single voxel TACs. A predicted path of the TACs was calculated using parameters R1 and BP$_{ND}$ estimated from the lp-ntPET analysis, yielding a fit that leaves out the compensatory functions thus representing the model fit as if no DA was released (for visualization purposes).

The following control analyses were performed: (1) the lp-ntPET model was fitted to TACs extracted from a priori defined striatal anatomical ROIs[49] to show that displacement was specific to associative striatum, mostly including caudate. (2) To investigate potential bias in the lp-ntPET model, a simulation analysis was performed (Supplementary Fig. 4a). (3) Each of the 5 gamma functions was fitted at 4 different time points (2 frames immediately surrounding the reversal onset and 2 frames later into the first reversal, thus including 20 models; Supplementary Fig. 4b). (4) An extension of the lp-ntPET model was used to fit all reversal events simultaneously (Supplementary Fig. 5). (5) Potential confounding variables were investigated in relation to the occupancy estimation (Supplementary Fig. 6a). (6) Single subject data for three representative subjects corresponding to low, medium, and high occupancy is shown in Supplementary Fig. 6b, c.

## fMRI

The BOLD fMRI data was acquired for 50 min starting 8 min after the PET scan start with the following parameters: FOV: 25.6, Matrix: $96 \times 96$, Slice Thickness: 3.6 mm, TE: 30 ms, TR: 4,000 ms, Flip Angle: 90°, Acceleration Factor: 2.0, resulting in a voxel size of $1.95 \times 1.95 \times 3.9$ mm3.

The fMRI data pre-processing was carried out using FEAT (FMRI Expert Analysis Tool) Version 6.00, part of FSL (FMRIB's Software Library, www.fmrib.ox.ac.uk/fsl). Registration of the functional data to the high resolution structural image was carried out using a boundary-based registration algorithm[50]. Registration of the high-resolution structural image to standard MNI152 space was carried out using FLIRT[46,51] and was then further refined using FNIRT nonlinear registration. The following pre-processing was applied: motion correction using MCFLIRT[46], B0 unwarping, slice-timing correction using Fourier-space time-series phase-shifting, non-brain removal using BET[52], spatial smoothing using a Gaussian kernel of FWHM 8 mm, grand-mean intensity normalization of the entire 4D dataset by a single multiplicative factor, highpass temporal filtering (Gaussian-weighted least-squares straight line fitting, with sigma = 25.0 s).

Due to the slow TR, it was not possible to disentangle the cue and choice part of the trial from the feedback; events were therefore defined as single whole trials. The first contrast of interest was defined around the first reversal as perseverance error (after first reversal) > rewarded correct response (25 trials before reversal). Perseverance error was defined as trials where a participant kept choosing the previously most rewarded choice before switching to the currently most rewarded choice. The contrast was investigated in a ROI defined from the PET analysis using FSL's randomise function and small volume correction with TFCE[53,54]. In the next analysis, reward, no reward, and unsigned RPEs (absRPE) at each trial were entered as regressors in a whole brain GLM with the absRPE covariate as regressor of interest. absRPE was orthogonalized with respect to the valence regressors, so that this analysis estimates voxels responding to absRPE, i.e. unexpectedness of a trial independent of the sign. The two GLMs are illustrated in supplementary Fig. 7a and b. Brain imaging results was visualized using Workbench View (Workbench Command 1.3.2).

### Reporting summary

Further information on research design is available in the Nature Portfolio Reporting Summary linked to this article.

## Data availability

The group- and individual level processed brain imaging data are available at https://zenodo.org/records/10100769 (https://doi.org/10.5281/zenodo.10100769). The behavioral data generated in this study are provided in the Source Data file. The unprocessed research data is available upon request from qualified researchers, provided that ethical and legal restrictions that govern data sharing are met. Participants in this study did not provide informed consent for public data sharing. Requests for data access should be directed to anna.rieckmann@unibw.de and will be dealt with promptly. Source data are provided with this paper.

## Code availability

Code to perform PET analysis as well as computational modelling of behavior is available at https://zenodo.org/records/10100769 (https://doi.org/10.5281/zenodo.10100769).

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

## Acknowledgements

We thank the PET/MR team at CancerCentrum and the Nuclear Medicine Department at Umeå University Hospital for their valuable help with data collection. Thanks Bertalan Polner for discussions about the data. This work was supported by the Swedish Research Council (grant 2015-03080) and the European Research Council under the European Union's Horizon 2020 research and innovation program (ERC-STG-716065 to A.R.). L.N. was supported by the Knut and Alice Wallenberg (KAW) foundation grant 2015.0277. M.G.M. was supported by a Project Grant (2021-02046) from the Swedish Research Council.

## Author contributions

Conceptualization: F.G., A.R., and L.N. Methodology: F.G., A.R., M.G.M., J.J., and J.A. Investigation: F.G., M.G.M., J.J., L.S., L.N., and A.R. Writing-original draft: F.G. Writing-review and editing: F.G., M.G.M., J.J., L.S., J.A., L.N., and A.R.

## Funding

## Competing interests

The authors declare no competing interests.
