## [Peer Review File · Nature Communications]

Dopamine release in human associative striatum during reversal learningReviewer #1 (Remarks to the Author):

Comments

In this study, Grill et al. use a two-armed bandit task with a stable and a volatile phase and record parallel BOLD-fMRI and dopamine (DA) release with PET, using raclopride which is sensitive to D2/D3 dopamine receptors. They investigate whether DA release is related to reward prediction errors (RPEs). They find PET DA and fMRI signals in the striatum at the time of the first reversal. They also show that the PET DA signal scales with interindividual differences in RPE sensitivity. The authors conclude that DA release is directly related to RPE processing during reversal learning.

This is a short paper with a simple message. It does not provide new mechanistic insights into the brain's reward system and in that sense, it is not going to change how we think about the brain. It does not test a new theory or hypothesis. Nevertheless, it provides direct in vivo evidence in humans that dopamine release is related to reward prediction errors. Please find a few comments below.

The choice of models seems unclear. For example, the authors do not consider some of the models used in previous related work such as one estimating the probability of a reversal (Bartolo & Averbeck, *Neuron*, 2020) or one that estimates the volatility of the environment and thus adjusts learning between stable and volatile environments (Behrens et al, *Nat Neurosci*, 2007). Also, the parameter recovery is not shown, and it is unclear why it is so low for the more complex model (i.e., which parameter is not recovering). More rigour and detail on the model selection and recovery would be helpful even if it has a small impact on the neural results.

One of the key analyses in Fig 2C relies on RPE magnitude/sensitivity estimates from a window of 25 trials. It is not clear how this window size was determined and whether the results are robust to the choice of the window size.

The BOLD responses seem rather small and unusual for an effect that is usually quite robust and replicable. There are a couple of things worth noting: First, it seems that the cue and outcome phase were not separable in their study which makes it hard to be sure that the BOLD responses are actual RPEs, rather than signals related to reward predictions and outcome (the two components of a prediction error). This is not acknowledged sufficiently or directly tested. Second, in the final analysis, the authors investigate unsigned RPEs. This is unusual given one of the most consistently reported features of striatal BOLD responses is that they signal signed RPEs (p.4, lines 155ff) and given the manuscript talks about RPEs as if they were signed. Relatedly, the authors describe that the reward and no reward regressors were included in the same model as the RPE, but these are not independent regressors and this probably only worked because the authors used the absolute RPE (a surprise signal). They should really run a model with (a) just RPE or (b) reward, no reward, and predicted reward, i.e. the components of the RPE and perform a model comparison (showing which of the two better explains the data in any given voxel). Finally, I don't think the authors analysed the DA release and BOLD signals when RPEs occurred in the stable phase in the absence of a reversal (i.e., incorrect trials in the stable phase which should also produce small RPEs). This seems essential to make a claim about DA being particularly relevant for RPEs around the reversal time where behavioural change occurs.

It is unclear how participants were instructed. Were they told that there were going to be changes, or did the instructions not mention the possibility of reversals? This question relates to the modelling as well (i.e., what priors participants might have had going into the experiment).

To make a claim about the absence of effects (Fig S1: NAc and Put, Fig S3), Bayesian tests are required.

Figure 2B is not very easy to read – maybe this could be split into two subpanels?

The reference numbering seems off.

Reviewer #2 (Remarks to the Author):

“Dopamine release in human associative striatum in response to reward-prediction errors during reversal learning: Evidence from functional hybrid PET-MR” Grill et al. Nature Communications. 10-02-22 Initial Submission.

In this paper the authors combine a reversal learning task, computational modeling, fMRI, and PET ([11C] Raclopride) to explore the relationship between dopamine release (PET) and the hemodynamic response (fMRI) in response to reward prediction errors.

Summary of the Main Results.

1. At a group level [11C] in relation to the reversal, Raclopride displacement, which is interpreted as DA release, was localized, to the caudate.

2. Individual subjects’ DA peak occupancy was negatively correlated with average RPE (reward prediction error) magnitude (as extracted from the behavioral model) for 25 trials after the first reversal. They were positively correlated with RPE sensitivity.

3. Confined to a striatal ROI defined by the PET imaging, there was significant BOLD activity for the contrast error post-reversal > reward pre-reversal.

4. A whole-brain analysis of “unsigned” (absolute value) RPE activation showed substantial overlap in the striatum with the PET activations (also in cortical regions).

Overall Impression

This is an interesting, solid study that relies on the different strengths (and timescales for inference) of fMRI and PET to provide further evidence for the relationship between RPEs and dopamine release in the striatum.

Further, individual parameters or signals (average RPE) from the computational model of behavior were related to individual PET activations, adding substantial strength to the study.

Overall the current contribution provides a very strong connection between three analysis levels - quantitative model-based behavioral estimation, PET results at the group level, and fMRI. The independence of the PET results and fMRI results allowed the authors to use one method to develop ROI hypotheses for the other.

There are just a few further issues to address.

Issues to address

1. The authors say the model was estimated in STAN. More details need to be given about the fitting, at least in the supplemental material. For example, was the model fit for each subject individually or was it a hierarchical model? Were the parameter estimates modes of the relevant posteriors? Also, it is stated the models were compared using a likelihood ratio test and reported in a supplementary table. The table states the likelihoods for the models but no details of the tests are given.

2. Line 125. TFCE is not defined yet.

3. Line 367. Typically, in the summary statistic approach to random effects analysis the parameter estimates are taken to the second level and subjected to statistical tests. Here it says first level t-tests were used at the second level. Is this correct?

Reviewer #3 (Remarks to the Author):

The manuscript titled "Dopamine release in human associative striatum in response to reward-prediction errors during reversal learning: Evidence from functional hybrid PET-MR" uses a combination of 11C-raclopride PET, fMRI and computational modeling of behavioral data to implicate dopamine release as a component of reversal learning. The multi-modal experiments described are challenging and well-executed with advanced analysis techniques previously published. However, a major criticism is that the authors did not appropriately address the question of bias in estimates of changes in 11C-raclopride BPnd when applying the lp-ntPET model. The main results of the study rest on this finding, so that this is a crucial piece of information to ensure that the presented results are accurate and robust. Overall, it is a well-constructed and very interesting study with a thoughtful experimental design that addresses a important questions. As it stands, it is in need for an appropriate approach or a negative control to validate their findings.

The following major points would need to be addressed in order to improve the quality of the manuscript:

1) Bias in kinetic model analysis of 11C-raclopride, especially with analyses of dynamic timecourses that model DA, can on the order of the changes in magnitude that the authors are observing. It is therefore crucial to ensure that the results are not driven by bias. The timing of the challenge occurs at ~38min after TOI, which allows for a robust baseline estimation. In a bolus design, this also means that the TAC is decreasing, so modeling an additional decrease (that is DA induced) can carry biases by nature of the analysis. Do the authors have any 11C-raclopride baseline data (without a task) and tried to run the same kinetic model on that data? Effectively, is there a negative control that the authors can check in order to answer the question of bias? A recent article (Levine et al. JCBFM 2022) discusses model bias in detail.

2) In the Methods, it seems that only 5 basis functions were used to fit the DA release timecourse? This seems very few, especially given the tight constraints of restricting the onset to -2 to +6min. How did the authors choose the timecourse/shape of the basis functions? The change in occupancy in Fig.2B does not seem to represent a natural DA response to a short Did the authors try to fit randomized timepoints in the TAC to ensure the validity of their approach?

3) The authors discuss that no 11C-raclopride displacement could be observed within the volatile periods. If a similar amount of DA release is expected for all transitions, this is somewhat worrying because 11C-raclopride should be capable of capturing occupancy changes on the order of 10-20% as the authors report for the first reversal. Radiotracer decay alone may not account for this. Similarly to the points above, fitting a model to several timepoints within a given TAC may not provide a stable estimate and that may be the reason the authors are not finding changes in occupancy.

4) Since 11C-raclopride has sensitivity only in the striatal areas, these are the only areas that can be explored with PET. However, to facilitate comparisons, BOLD contrast for perseverance error was restricted to look at effectively only PET ROIs. This makes sense to do as a first comparison. From a neuroscience perspective, it is not only interesting how overlapping anatomical regions correlated, but also how the whole brain response looks like.

5) In the discussion, the authors state that there was a temporal and spatial overlap with the DA release cluster. This statement may not be entirely appropriate. The authors

do not report temporal BOLD data, or a temporal correlation. In terms of spatial overlap, there is some spatial overlap but there are also areas that do not show BOLD contrast but do show DA release. How do the authors interpret that finding?

6) It would be informative to report time activity curves, plus model fits. This could be done as an average, or from some representative sample subjects. While this is not common in fMRI, partly because analysis methods are much more established, it is of importance in PET and the application of newer, less well-validated models.

7) An extended discussion on the DA timecourses and expected magnitude would benefit this manuscript. Are the reported values from this study in line with e.g. those from animal studies? 10-20% increases are quite large for a behavioral challenge, and almost mimic those from some drug-induced changes. How do the authors explain that?

We are pleased that all reviewers see merit in our work. In the revised version, we have tried to address all the thoughtful comments and suggestions for clarifications and new analyses. Below, we include a point-by-point response to the issues raised by each reviewer. The corresponding changes in the main manuscript and supplement are highlighted in blue in all documents. Please note that in response to reviewer 2, we re-analyzed the entire set of PET data and all related fMRI data, so all figures and statistics have been updated throughout the manuscript (these are not highlighted at every instance).

Reviewer #1 (Remarks to the Author):

Comments

In this study, Grill et al. use a two-armed bandit task with a stable and a volatile phase and record parallel BOLD-fMRI and dopamine (DA) release with PET, using raclopride which is sensitive to D2/D3 dopamine receptors. They investigate whether DA release is related to reward prediction errors (RPEs). They find PET DA and fMRI signals in the striatum at the time of the first reversal. They also show that the PET DA signal scales with interindividual differences in RPE sensitivity. The authors conclude that DA release is directly related to RPE processing during reversal learning.

This is a short paper with a simple message. It does not provide new mechanistic insights into the brain's reward system and in that sense, it is not going to change how we think about the brain. It does not test a new theory or hypothesis. Nevertheless, it provides direct *in vivo* evidence in humans that dopamine release is related to reward prediction errors. Please find a few comments below.

In response to this overall comment by Reviewer 1, we like to stress that our *in vivo* hybrid imaging work supports a model of human reversal learning in which dopaminergic responses to reward prediction errors activate the human mesocortical dopamine system, separable from reward-coding neuronal populations in ventral striatum, which together with recent animal work (e.g. Ishino et al. 2023; (<https://doi.org/10.1126/sciadv.ade5420>), goes well beyond the traditional framework. Moreover, unlike prior theories that have focused on individual differences in tonic dopamine levels, our work shows for the first time that it is the *reactivity* of the human dopamine system and not baseline levels of dopamine that underlies flexibility in reversal learning. Through quantitative measurements of neurotransmitter release, we show that larger amounts of dopamine release are highly proportional to a faster reversal of behavior when learned associations change. We also establish that neither rigid nor overly flexible learning is best for task success, suggesting a medium amount of dopamine release as optimal. Further, and in contrast to prior work with pharmacological manipulations or genetics, our human *in vivo* imaging work allows us to *spatially locate dopamine release* during reversal learning to the dorsal striatum, with corresponding fMRI activity in cortical cognitive-control regions. We have added some remarks to the discussion section of the revision (line 219) and hope these will better convey the novelty of the contribution.

1. The choice of models seems unclear. For example, the authors do not consider some of the models used in previous related work such as one estimating the probability of a reversal (Bartolo &

Averbeck, Neuron, 2020) or one that estimates the volatility of the environment and thus adjusts learning between stable and volatile environments (Behrens et al, Nat Neurosci, 2007). Also, the parameter recovery is not shown, and it is unclear why it is so low for the more complex model (i.e., which parameter is not recovering). More rigour and detail on the model selection and recovery would be helpful even if it has a small impact on the neural results.

We agree with the reviewer that rigor for model selection and recovery is important and have updated the manuscript methods, discussion and supplementary accordingly. The purpose of using RL models to explain the behavioral data was to be able to estimate trial-by-trial RPEs that would then be linked to the fMRI data through a RPE regressor and to the PET data by averaging the RPE magnitude over the critical reversal. In the revised methods section, we describe the model selection more carefully.

The additional model proposed by the reviewer (Bartolo & Averbeck (2020)) is of great importance to the general field of reversal learning. An important distinction between their and our task paradigm is that they overtrained the subjects so that they acquired a Bayesian model of the task (including reversals) which they used to make choices. In our task, we did not inform the participants that there are any reversals in the task (which we reasoned would maximize RPEs at the reversals). That is, our participants had not acquired a model that included reversals at the timepoint where we observe dopamine release. Therefore, we do not consider the model by Bartolo & Averbeck (2020) appropriate to fit to our data. It is however entirely possible that the observed dopamine release reflects the initial updating of a model that includes reversals. We would like to thank the reviewer for pointing out this literature since it made us think about the results from a different angle and we have added this to the discussion section.

Changes made:

- We have included the parameter recovery analysis (true vs. predicted parameter correlations) in the supplementary material (Supplementary Fig. 3).
- Line 235 added: “Since participants were not instructed that the task contained reversals, it is possible that the observed DA release reflects updating of an internal task model which include reversal monitoring. This view is congruent with our exploratory analyses finding no evidence for DA release of a similar magnitude and reliability during the volatile period (Supplementary Fig. 5).”.
- Line 279 added: “Previous research on reversal learning has emphasized the modulation of RPE sensitivity when transitioning from stable to volatile environments^{12, 39}. In an attempt to address such modulation we fit a previously established model that allowed RPE sensitivity to be modulated trial by trial depending on choice confidence⁴⁰. However, this model did not outperform a simple reinforcement learning model which estimates a single RPE sensitivity using all trials.”
- Methods were updated throughout lines 346-415 to clarify our basis for model selection.

2. One of the key analyses in Fig 2C relies on RPE magnitude/sensitivity estimates from a window of 25 trials. It is not clear how this window size was determined and whether the results are robust to the choice of the window size.

Please note that it is only RPE magnitude that is calculated using an average of the 25-trial window after the first reversal. RPE sensitivity is estimated for the whole task so all trials are taken into account when estimating this parameter using the computational model. We have clarified this in the figure 2 legend: “Peak DA receptor occupancy was correlated with mean RPE magnitude over a 25 trial windows after the first reversal and with RPE sensitivity (note that RPE sensitivity was estimated on all trials).”

The window size corresponds to the entire first reversal block. Thus, it was consistent with the onset of observed dopamine release, including all subsequent trials and based on the task design. Using an arbitrarily larger window size would include the next reversal where the RPE magnitude again increases but where we could not estimate dopamine release onset reliably. Thus, in our opinion, increasing the window size arbitrarily reduces the temporal correspondence of the comparison as well as its relation to the task structure. In response to the reviewer’s concern about robustness, we have repeated the same correlational analysis with a window size varying between 5 and 25 trials (see below). This shows that the result is robust for window sizes > 6. This figure was added as Supplementary Fig. 2b.

3. The BOLD responses seem rather small and unusual for an effect that is usually quite robust and replicable. There are a couple of things worth noting: First, it seems that the cue and outcome phase were not separable in their study which makes it hard to be sure that the BOLD responses are actual RPEs, rather than signals related to reward predictions and outcome (the two components of a prediction error). This is not acknowledged sufficiently or directly tested. Second, in the final analysis, the authors investigate unsigned RPEs. This is unusual given one of the most consistently reported features of striatal BOLD responses is that they signal signed RPEs (p.4, lines 155ff) and given the manuscript talks about RPEs as if they were signed. Relatedly, the authors describe that the reward and no reward regressors were included in the same model as the RPE, but these are not independent regressors and this probably only worked because the authors used the absolute RPE (a surprise signal). They should really run a model with (a) just RPE or (b) reward, no reward, and predicted reward, i.e. the components of the RPE and perform a model comparison (showing which of the two better explains the data in any given voxel). Finally, I don’t think the authors analysed the DA release and BOLD signals when RPEs occurred in the stable phase in the absence of a reversal (i.e., incorrect trials in the stable phase which should also produce small RPEs). This seems essential to make a claim about DA being particularly relevant for RPEs around the reversal time where behavioural change occurs.

We thank the reviewer for this detailed comment and respond below to the different concerns.

a. “it seems that the cue and outcome phase were not separable in their study which makes it hard to be sure that the BOLD responses are actual RPEs, rather than signals related to reward predictions and outcome (the two components of a prediction error).”

Indeed, this is a limitation in the current study due to the tradeoff necessitated by collecting fMRI and PET simultaneously in a single behavioral paradigm. If only fMRI was used, a better jitter between cue and outcome could have better disentangled the various components. However, with both an ISI and ITI jitter we reasoned that the PET part of the experiment would have a diminished chance of success since the RPEs at the reversals would occur at a much slower and more variable rate. We have included this consideration in the limitations section line 283: “Finally, future studies should consider a more optimal jitter between cue and outcome for each trial for optimal separation of these events.”

b. “The authors describe that the reward and no reward regressors were included in the same model as the RPE, but these are not independent regressors and this probably only worked because the authors used the absolute RPE (a surprise signal). They should really run a model with (a) just RPE or (b) reward, no reward, and predicted reward, i.e. the components of the RPE and perform a model comparison (showing which of the two better explains the data in any given voxel).”

The experimental paradigm is set up to produce *trains of negative RPEs* especially at the first reversal and the concept of lack of expected reward as a trigger of reversal learning is a central aspect of the study. During the first 30 minutes of the task, participants learn that the index finger is the most rewarded choice. Due to the probabilistic setup of the paradigm, there will be some negative RPEs during this stable period around 20% of the time even after the initial learning, i.e., the expected value and actual reward will be congruent 80% of the trials. Unbeknownst to the participants, the reward contingencies switch and there will be a train of trials where the negative RPE (perseverance errors) magnitude is high until a behavioral shift. As we show in Fig. 1c, the RPE magnitude (absolute/unsigned RPE) therefore reflects, for a large majority of trials, negative RPEs. Accordingly, our primary fMRI model compares fMRI responses to perseverance errors after the reversal to rewarded correct responses immediately before the reversal. Nevertheless, even though this is the model that most closely aligns with the PET model, this model is not able to disentangle surprise components of negative prediction errors from a change in valence. Therefore, we fit a second fMRI model that isolates the surprise component of reward predictions from valence on a trial-by-trial basis throughout the entire run, i.e. not selective to the reversal. We reasoned that if striatal activation in both models spatially aligns with dopamine release, it supports an interpretation of dopamine release to lack of expected reward through surprise signals rather than through a change in reward rate even though we are not able to capture surprise *per se* in the PET model or this task design. This fMRI model was set up to account for the valence part as a regressor of no interest (reward vs no-reward) and separate this from trial-by-trial absRPE, i.e., unexpected/surprise signal magnitude beyond valence (by design, mainly negative). We find that both fMRI contrasts activate dorsal striatal areas and overlap with the location of dopamine release. In addition, we find that the valence contrast (Supplementary Fig. 7) captures canonical reward areas including ventral striatum but that violations of expected rewards elicit activation in dorsal striatum and, crucially, that this is also paralleled by dopamine release in dorsal striatum. We have revised Supplementary Fig. 7 to better show this distinction of DA release as a “teaching signal” of reversal learning from signals in canonical ventral reward areas.

Thus, we agree with the reviewer that “these are not independent regressors and this probably only worked because the authors used the absolute RPE (a surprise signal)” but rather than seeing this as a limitation, we did it on purpose in this second model such that each voxel’s activity correspond to either valence or RPE in the second fMRI model.

Research has shown that DA is released in the striatum to unexpected events (i.e. irrespective of stimuli valence e.g. Clarke 2011, Hamid 2021, Hart 2014). Also, recent research by Ishino et al. 2023 (<https://doi.org/10.1126/sciadv.ade5420>) has shown that certain DA neurons in the midbrain of rodents fire in a way that is orthogonal to what was reported by Schultz et.al. 1997, projecting to the dorsal parts of the nucleus accumbens (near the border of caudate), and Ishino et al show that DA is released in this striatal area together with the midbrain firing. Even though these data are from invasive recordings at a much faster timescale than in our study, the results are highly consistent with what we observe in terms of DA release and BOLD representation of unsigned RPEs in the striatum.

Changes made:

- **Introduction** line 46: “ While dopamine release in response to RPEs has traditionally focused on unexpected positive outcomes, recent animal work has shown that DA is released to unexpected events irrespective of stimuli valence⁶⁻⁹. Moreover, Ishino et al. found that DA neurons in the midbrain of rodents that project to the dorsal parts of the nucleus accumbens (near the border of caudate) signal learning from lack of expected rewards⁹. “
- **Results** line 151: BOLD responses: Due to the temporal resolution of PET, the identification of DA release to RPEs was confined to the reversal period. To capture concurrent BOLD activation at the first reversal, from stable to volatile task period, a primary BOLD contrast of interest was defined to compare rewarded correct responses on the 25 trials before the first reversal with perseverance errors over 25 trials after the first reversal. In order to focus on voxels that overlap with the location of DA release, the DA release cluster was used as a mask. Significant BOLD response differences were observed in this cluster in the right caudate (peak MNI-coordinate = xyz[12,10,0], $t(25) = 2.99$, $k\text{-voxels} = 14$, $p < 0.05$, TFCE corrected; Fig. 3a). A second analysis identified hemodynamic responses from fMRI to trial-level unsigned RPEs encountered over the entire experiment and in whole brain. RPEs were coded on the single trial level according to the computational model and orthogonalized with respect to valence of each trial. We reasoned that overlap of this second model with DA release supports a conclusion that links DA release to lack of rewards at reversal with the surprise component of RPEs more generally rather than to a change in valence.
- **Methods** line 489: “In the next analysis, reward, neutral, and unsigned RPEs at each trial were entered as regressors in a whole brain GLM with the RPE covariate as regressor of interest. RPE was orthogonalized with respect to the valence regressors, so that this analysis estimates voxels responding to RPE, over and above its shared variance with valence.”
- Revised Supplementary Fig. 7

c. “ I don’t think the authors analysed the DA release and BOLD signals when RPEs occurred in the stable phase in the absence of a reversal (i.e., incorrect trials in the stable phase which should also produce small RPEs). This seems essential to make a claim about DA being particularly relevant for RPEs around the reversal time where behavioural change occurs.”

We agree that it would be extremely valuable to be able to model PET and fMRI data congruently on the single trial level during the stable phase in the absence of a reversal. But in our view, DA release cannot be estimated during the stable task period since there is no baseline to compare it to and single-trial analysis is impossible due to the kinetics. Nevertheless, we do agree with the reviewers that our results should, in theory, also hold for the small RPEs in the stable phase. This is the reason for including two fMRI models in the manuscript: (1) time locked with the PET data, experimentally induced by reversals [i.e. comparing 25 trials before the first reversal and 25 trials after the first reversal (perseverance error contrast), Figure 3A), and (2) taking all trials into account using RPEs from the computational models, i.e. including the small RPEs in the absence of the reversal (Figure 3B). The black outline shows that both analyses overlap with the site of DA release, which we take as (imperfect) evidence that our conclusions generalize to RPEs encountered in the absence of the experimental reversal. We clarify this reasoning in line 164, in addition to the changes made to the previous comment: *“Moreover, because this second model was fit trial-by-trial, temporally removed from the PET model, overlap between DA release and activation in this second model would speak against a link between DA release and slow contextual shift in volatility.”*

4. It is unclear how participants were instructed. Were they told that there were going to be changes, or did the instructions not mention the possibility of reversals? This question relates to the modelling as well (i.e., what priors participants might have had going into the experiment).

We agree that this is an important aspect of the study and have clarified in the manuscript that the participants were not instructed that there was a possibility for reversals. Thus, concerning priors, the participants were not actively monitoring any change-points based on their instructions (please see also our response to your comment 1).

5. To make a claim about the absence of effects (Fig S1: NAc and Put, Fig S3), Bayesian tests are required.

We have indeed now computed the analysis using Bayesian statistics and report Bayes factors where we make any claims on null effects.

6. Figure 2B is not very easy to read – maybe this could be split into two subpanels?

Thank you for this suggestion. We have edited Figure 2b (contents now in Fig. 2c) in an attempt to make it more readable. We show the average PET data and model fit over the group in one panel and in a second panel we show the estimated displacement functions that give rise to the “dip” in the ROI time activity curve. We hope this improves readability of the figure -- it is a central aspect of the study.

7. The reference numbering seems off.

Thank you. We have double checked the reference numbering and indeed found some that were off.

Reviewer #2 (Remarks to the Author)

This is an interesting, solid study that relies on the different strengths (and timescales for inference) of fMRI and PET to provide further evidence for the relationship between RPEs and dopamine release in the striatum. Further, individual parameters or signals (average RPE) from the computational model of behavior were related to individual PET activations, adding substantial strength to the study. Overall the current contribution provides a very strong connection between three analysis levels - quantitative model-based behavioral estimation, PET results at the group level, and fMRI. The independence of the PET results and fMRI results allowed the authors to use one method to develop ROI hypotheses for the other.

Issues to address

1. The authors say the model was estimated in STAN. More details need to be given about the fitting, at least in the supplemental material. For example, was the model fit for each subject individually or was it a hierarchical model? Were the parameter estimates modes of the relevant posteriors? Also, it is stated the models were compared using a likelihood ratio test and reported in a supplementary table. The table states the likelihoods for the models but no details of the tests are given.

More details regarding the behavioral model fitting have been added to the supplementary material: “Behavioral Model fitting: The behavioral data were fit using hierarchical models with hyperpriors for each group-level free parameter’s mean and variance. The hyperpriors were weakly informed using a normal distribution with a mean of 0 and standard deviation of 1. The individual level free parameters were given the same priors. STAN in R4.1.1 was used to fit the data with 4 chains, 6000 iterations (5000 warmup), yielding 4000 posterior estimates for each parameter and individual, the mean of the posterior estimates were used in the parameter recovery analysis and the correlation analysis (RPE sensitivity (\$\alpha\$ ) and peak dopamine occupancy; Main Manuscript Fig. 2c).

Regarding the likelihood ratio test: This is an oversight on our part that we have clarified. Models were initially chosen using minimum likelihood estimates but this will of course bias the decision towards more complicated models since number of model parameters are not taken into account. We now report the raw likelihood estimation as well as Akaike information criterion in the tables which takes number of model parameters into account. Based on other review comments we have also added considerable information regarding model selection and model/parameter recovery (Supplementary Table 2 and Supplementary Fig. 3).

Line 98: “To estimate RPE magnitude and associated parameters related to task performance, we fitted a series of reinforcement learning models to the observed behavioral data and assessed which model best accounted for the data by comparing likelihood estimates as well as through the Akaike information criterion (AIC; Supplementary Table 1). The initial winning model with lowest likelihood estimation included a set of parameters that dynamically modulated the learning rate parameter α (RPE sensitivity) from trial to trial. However, this model had a higher AIC estimate than a more simple model, and only two of three of the fitted parameters were able to be recovered (α_0 : $r = 0.83$; β : $r = 0.83$; κ : $r = -$

0.12). The model recovery analysis showed that a simpler model was better able to account for data simulated from the initially winning model (Supplementary Table 2). Due to these observations, the simpler model (Model 1) was chosen for the subsequent analyses. Large individual differences were observed in the fitted parameters (α : M = 0.38, SD = 0.26; β : M = 3.20, SD = 4.64).”

2. Line 125. TFCE is not defined yet.

Thank you, TFCE is now correctly defined as threshold free cluster enhancement at this place in the text.

3. Line 367. Typically, in the summary statistic approach to random effects analysis the parameter estimates are taken to the second level and subjected to statistical tests. Here it says first level t-tests were used at the second level. Is this correct?

We would like to thank the reviewer for this comment as the first level t-statistics that was used in the group level is a mistake on our part. The first level statistics should be the “g-statistic” (similar to beta in regression models). We have corrected this mistake in the revised manuscript and now use “parameter estimate maps” instead of t-statistic maps.

Reviewer #3 (Remarks to the Author):

The manuscript titled “Dopamine release in human associative striatum in response to reward-prediction errors during reversal learning: Evidence from functional hybrid PET-MR” uses a combination of 11C-raclopride PET, fMRI and computational modeling of behavioral data to implicate dopamine release as a component of reversal learning. The multi-modal experiments described are challenging and well-executed with advanced analysis techniques previously published. However, a major criticism is that the authors did not appropriately address the question of bias in estimates of changes in 11C-raclopride BPnd when applying the lp-ntPET model. The main results of the study rest on this finding, so that this is a crucial piece of information to ensure that the presented results are accurate and robust. Overall, it is a well-constructed and very interesting study with a thoughtful experimental design that addresses a important questions. As it stands, it is in need for an appropriate approach or a negative control to validate their findings.

The following major points would need to be addressed in order to improve the quality of the manuscript:

1) Bias in kinetic model analysis of 11C-raclopride, especially with analyses of dynamic timecourses that model DA, can on the order of the changes in magnitude that the authors are observing. It is therefore crucial to ensure that the results are not driven by bias. The timing of the challenge occurs at ~38min after TOI, which allows for a robust baseline estimation. In a bolus design, this also means that the TAC is decreasing, so modeling an additional decrease (that is DA induced) can carry biases by nature of the analysis. Do the authors have any 11C-raclopride baseline data (without a task) and tried to run the same kinetic model on that data? Effectively, is there a negative control that the

authors can check in order to answer the question of bias? A recent article (Levine et al. JCBFM 2022) discusses model bias in detail.

We thank the reviewer for this thoughtful comment and acknowledge that this type of model can be susceptible to inherent noise in the data which might bias the estimations. We understand the concern by the reviewer that it is a possibility that the DA release we see is an artifact of the analysis and design or, in other words, a false positive. In response to the reviewer's comment, we would like to make several points that, at least for us, have provided evidence that our results are not reflecting bias in the kinetic model.

- We do not have real “negative control” – task-free data. It was a deliberate design choice to not use resting state as a baseline or to compare our results to conditions with little experimental control over the subject's cognitive state and processes. We do believe that our active task baseline is a superior way to set up DA release behavioral paradigms compared to using a resting baseline since the displacement will be specific to the component process of a task (here, reversing probabilities) rather than reflecting widespread general DA release when comparing rest to task. While we do acknowledge a risk for bias in the lp-nt-PET (Levine et al, 2022), we think the fact that the site of DA release is so clearly localized to the associative striatum, overlaps with fMRI activity, and that individual estimates of release were highly correlated with task performance, make it unlikely that our conclusions are influenced by methodological confounds.
- In a bolus design, the TAC is indeed decreasing over time, but the lp-ntPET can be thought of as an extension of the MRTM. That is, the cerebellum is used as a reference region which accounts for the blood compartment of the model. If any decrease in the signal of the TAC of interest co-occurs with the blood compartment, this will not be modelled as Raclopride displacement. Hence, as is standard in a kinetic model, the decreasing nature of a bolus design is accounted for here as well.
- In the revised manuscript, we have added single subject data for 3 representative individuals that display low (5.7%), medium (11.1%), and high (19.9%) occupancy (Supplementary Fig. 6a). Supplementary Fig. 6b shows the corresponding voxelwise parametric maps from the first level lp-ntPET analysis. Note that the parameters can be negative since we allow the model to fit “positive displacement” (which is most likely noise) to not bias our results to only be able to capture decreases in the TAC. Also, we are not restricting the voxelwise analysis to the striatum, which can be seen from some positive voxels in the thalamus and some negative voxels in the occipital cortex. We did this to be open to bias in the model fit. To further investigate any bias in our data we looked at associations between the estimated occupancy and potential confounds over the whole group. For example, a potential confound relates to the uptake of the radio tracer, where one could imagine that lower uptake leads to noisier data, which potentially leads to a bias in estimating occupancy. We therefore correlated occupancy with max radioactivity count in the ROI TAC and the reference region TAC and found no significant association. Another potential confound relates to motion of the subjects, as a potential bias in the estimation of occupancy can occur if participants move during the scan. No significant association was observed between motion and occupancy (Figure S6C, below).

Added text to limitations line 271: There are also some limitations to the current study. Recent work has pointed out a number of potential methodological problems for single-scan designs in PET³⁸ (Levine et al. 2022). We found no confounding effects of head motion and adding control functions in the PET model (that fit DA release incongruent with the reversal) did not alter the spatial statistics. Further, the fact that the site of DA release is clearly localized to the associative striatum, overlaps with fMRI activity, and correlated with performance on the task, it is unlikely that the conclusions we draw are driven by methodological confounds. However, a trade-off induced by our controlled experimental design is that we have no task-free resting state PET data to quantify model bias in our set-up.

- To further investigate potential bias in our modelling approach we simulated TACs with the range of noise estimated from the residuals of the MRTM fit. We then fitted the lp-ntPET model on the simulated TACs, and since no activation was included in the simulation any significant displacement estimated by the model is considered a false positive due to noise. We found 4.86% false positives which indicates that the model is not over-estimating displacement due to noise, at our chosen alpha of 0.05. We have added this analysis to the supplementary materials, Supplementary Fig. 4a. Please find added supplementary figures below.

2) In the Methods, it seems that only 5 basis functions were used to fit the DA release timecourse? This seems very few, especially given the tight constraints of restricting the onset to -2 to +6min. How did the authors choose the timecourse/shape of the basis functions? The change in occupancy in Fig.2B does not seem to represent a natural DA response to a short Did the authors try to fit randomized timepoints in the TAC to ensure the validity of their approach?

We apologize that we did not accurately describe the selection of basis functions. We actually fitted different models and have clarified this now. In all models, we use some weakly informed priors regarding shape (gamma function) and onset (using the experimental paradigm to focus in on the onset of the experimental manipulation). The main model with 5 functions fits functions congruent with the experimental manipulation at the frame onset closest to the reversal. A second model, added to ensure validity along the lines of the reviewer's suggestion, fits these functions at the same frame plus, simultaneously, 3 additional frames after the reversal onset (i.e., 20 functions). The voxel statistics for these two models are highly similar (see below and in new supplement material and Supplementary Fig. 4b), providing evidence that "non-congruent onsets" after the reversal do not capture any additional displacements of the TAC. Also note that both models allow functions to be fitted in both directions (where positive diversions from the TAC again reflect a non-sensical result that we do not observe in this main cluster. A supplementary figure with individual data in Supplementary Fig. 6b shows that these are sometimes captured on the single subject level in regions outside the striatum). A final, exploratory,

model fitted also functions that show multiple displacements in line with the multiple reversals during the volatile phase. More detail on this has been added to supplement and Supplementary Fig. 5 (see next point).

3) The authors discuss that no ¹¹C-raclopride displacement could be observed within the volatile periods. If a similar amount of DA release is expected for all transitions, this is somewhat worrying because ¹¹C-raclopride should be capable of capturing occupancy changes on the order of 10-20% as the authors report for the first reversal. Radiotracer decay alone may not account for this. Similarly to the points above, fitting a model to several timepoints within a given TAC may not provide a stable estimate and that may be the reason the authors are not finding changes in occupancy.

We agree with the reviewer and radiotracer decay was just mentioned as one possible problem that might influence detection of release at late stages of the task. We do not expect similar amounts of DA release during all transitions since the switch from a long stable to volatile phase at the first reversal is different than switches within the volatile phase. We have clarified in the text that our most likely explanation for DA release at the first reversal is reflective of the cognitive processes during the first reversal. We have moved the alternative, more methodological, explanations to the supplementary figure legend.

- Line 229: “In line with our observation of low RPE magnitude after reversal being coupled with high DA release, it is possible that the observed DA release reflects a higher order cognitive-control process that is influenced by RPEs rather than RPE per se. The theory of opportunity cost models asserts that diminishing cognitive control occurs when the average reward rates are high²⁹. Conversely, decreasing the average reward rate, as in the transition between stable and volatile period utilized in this study, should increase cognitive control³⁰, which has been linked to DA functions³¹. Since participants were not instructed that the task contained reversals, it is possible that the observed DA release reflects updating of an internal task model to include reversal monitoring. This view is congruent with our exploratory analyses finding no evidence

for DA release of a similar magnitude and reliability during the volatile period (Supplementary Fig. 5)”

- Supplement: “To investigate DA release for the consecutive reversals, a 1p-ntPET model was fit to the TACs of the significant DA release ROI for each participant that included basis functions that covered these events (Fig. 5a). The model seeks to fit the best possible combinations of basis functions to explain the data yielding a continuous estimation of DA occupancy (Fig. 5b). On the mean level over the group, the estimation looks reasonable. However, the individual level functions show negative occupancy values for some participants at the later reversals as well as some participants where a basis function only to the first reversal best explained the data. For these reasons, only the first reversal event was considered to yield a reliable estimation and was used for the results in the main manuscript. As discussed in the main text, this might reflect a specific cognitive process occurring at the first reversal, but could also, at least in part, be due to the lower radioactivity and slow pharmacokinetics of [11C]Raclopride during these later time points.”

4) Since 11C-raclopride has sensitivity only in the striatal areas, these are the only areas that can be explored with PET. However, to facilitate comparisons, BOLD contrast for perseverance error was restricted to look at effectively only PET ROIs. This makes sense to do as a first comparison. From a neuroscience perspective, it is not only interesting how overlapping anatomical regions correlated, but also how the whole brain response looks like.

We agree with the reviewer that it is potentially interesting to show fMRI responses outside the PET ROI as well, but please note that the perseverance error contrast relies only on a limited subset of trials and is therefore not ideally powered to detect small whole-brain effects. Fig. 3a has been edited to include also the t-statistics from the whole brain as an underlay. This does indeed provide some potentially interesting information such as a relatively strong midbrain response seen below. However, due to the uncorrected threshold we are hesitant to further interpret these effects here. Please also note that in the much better powered analysis of all RPE trials (model 2 in the fMRI part), whole brain results are presented in revised Supplementary Fig. 7 in response to reviewer 1 and discussed in the text.

5) In the discussion, the authors state that there was a temporal and spatial overlap with the DA release cluster. This statement may not be entirely appropriate. The authors do not report temporal BOLD data, or a temporal correlation. In terms of spatial overlap, there is some spatial overlap but there are also areas that do not show BOLD contrast but do show DA release. How do the authors interpret that finding?

We understand the reviewer's argument and have revised the wording regarding temporal overlap since they do indeed operate on different timescales. Our intention was to convey the finding that we find (partial) spatial overlap for independent modalities that are modeled with respect to the same time-locked experimental manipulation (the reversal). We do see that this is not the same as a true temporal overlap (see also reviewer 3, comment 3c). We agree that it is interesting that there are voxels with significant DA release with no corresponding significant BOLD activation, though an absence of overlap can be due to many reasons. We have added a paragraph about potential interpretations of this finding.

Changes made:

- Abstract: The fMRI hemodynamic response to perseverance errors at the onset of a reversal spatially overlapped with the site of dopamine release.
- Line 178: "In this study, we used simultaneously acquired dynamic [11C]Raclopride PET-fMRI and computational modeling to provide evidence for striatal DA release to lack of expected

reward as a core component of reversal learning in humans. We detected significant DA release in associative striatum, congruent with a reward reversal and consistent with recent animal work that has established DAergic responses as a teaching signal of learning. BOLD activations to perseverance errors that coincide with the reversal were identified in the location of DA release in associative striatum. An additional analysis of fMRI data to event-related RPE throughout the task also shows that RPE responses per se, not necessarily linked to a reversal period, are spatially consistent with the site of DA release and separable from a canonical reward response in ventral striatum.”

- Line 242: “The BOLD response to perseverance errors during the first reversal manipulation showed a spatial overlap with the striatal DA release cluster fitted at the same timepoint in the design. Extending the analysis beyond the reversal period with a trial-by-trial RPE regressor over the whole task revealed that hemodynamic RPE responses are also spatially congruent with the DA release cluster and extended to right anterior insula, bilateral parietal cortex, dorsolateral prefrontal cortex (DLPFC).”
- Line 256: “What remains unclear from this human PET modeling work is whether and how DA PET signals and the BOLD signal are mechanistically linked, both where they overlap and where they do not overlap. A dose response relationship between raclopride and cerebral blood volume has previously been observed in non-human primates, providing ample evidence for neurovascular coupling of DA binding to D2/D3 receptors³⁷. However, using a ligand specific to D2/D3 receptors in humans with a cognitive task instead of selective pharmacological manipulations, we cannot conclusively speak to the influence of DA release on the BOLD signal since this is dependent on what type of receptor (D1-like or D2-like) DA binds to; in the current study it is likely that DA binds to a mix of receptor types. Moreover, also non-DAergic signals are captured in the BOLD signal. Thus, BOLD activations that are not accompanied by DA release might indicate additional neurotransmitter processes within a cortico-striatal network. Ideally, future hybrid PET-MR studies would implement multi-tracer studies or combine cognitive with pharmacological challenges in order to capitalize on the unique possibility of hybrid PET-fMRI to understand network-wide brain dynamic in terms of underlying neurotransmitter action..”

6) It would be informative to report time activity curves, plus model fits. This could be done as an average, or from some representative sample subjects. While this is not common in fMRI, partly because analysis methods are much more established, it is of importance in PET and the application of newer, less well-validated models.

Thank you, this is a very good point, and we have edited and added several additional figures to the manuscript and to the supplementary materials showing this. First Fig. 2c (previously Fig. 2b) now shows the average time activity curve, reference time activity curve, lp-ntPET model fit, and the predicted path of the time activity curve (i.e., what the curve is predicted to look like if we did not account for Raclopride displacement using the activation functions). Additionally, the figure shows the average and individual dopamine occupancy profiles calculated from the activation functions. In the supplementary material (Supplementary Fig. 6a) we show single subject PET data and model fits from three individuals

corresponding to low (5.7 %), medium (11.1%), and high (19.9%) occupancy. The data from Supplementary Fig. 6a was extracted from the significant group level ROI (Main Manuscript Fig. 2a).

7) An extended discussion on the DA timecourses and expected magnitude would benefit this manuscript. Are the reported values from this study in line with e.g. those from animal studies? 10-20% increases are quite large for a behavioral challenge, and almost mimic those from some drug-induced changes. How do the authors explain that?

To our knowledge, it is not entirely clear what range of percentage increase in DA one can expect to find in humans for a behavioral challenge, but this is an interesting point for the discussion, and we have added the following text:

Line 200: “In sum, through quantitative measurements of neurotransmitter release, we show that larger amounts of DA release are highly proportional to a faster reversal of behavior when learned associations change. Our work shows that it is the reactivity of the human DA system (i.e., the release in response to a specific change in error probability) and not level of DA receptor density that underlies flexibility in reversal learning. We also establish that neither rigid nor overly flexible learning is best for task success, suggesting a medium amount of dopamine release as optimal. That said, from our study it remains unclear what a medium amount of DA release generally constitutes and whether the magnitude is dependent on experimental design. Comparisons across studies suggest that our mean change in BP_{ND} of 12.65 % is higher than the 5-10 % estimates from PET studies of striatum looking at rewarding stimuli (e.g.^{18,27}), though on par with a study on conditioning (e.g.¹⁹). How these estimates are comparable and what constitutes a normal amount of DA release remains to be determined by studies that combine pharmacological control and task situations in the same design. Of note, a study by Boileau et al. showed comparable amounts of DA release either in response to amphetamine or to a (conditioned) placebo¹⁷, suggesting that a relatively strong DA release in the context of our learning study appears plausible and is consistent with a general role for DA in learning about rewards rather than reward per se.”

Reviewer #1 (Remarks to the Author):

The authors have done a great job addressing most of my comments and I believe the manuscript has become clearer and stronger as a result. I have a few outstanding comments that have not been fully addressed.

(1) First, I am puzzled about why the authors prefer to stick to the terminology of a prediction error. Both, reinforcement learning theory by Sutton and Barto (and others) as well as the seminal findings by Schultz and colleagues in DA neurons (and many others since then), postulate that the prediction error is a signed response (i.e., positive response of DA neurons to outcomes that are more positive than expected and suppression of DA response to outcomes that are more negative than expected). I appreciate that other views are interesting and valid, but maybe using the same terminology of a RPE is then misleading because readers will assume the authors are talking about a signed response. Instead, here, the authors are focusing on an absolute signal which they call RPE amplitude. I suggest that the terminology is adapted so that it is intuitive for the reader and consistent with the literature. RPE magnitude is the same as surprise or absolute RPE which would be more appropriate. A magnitude could be signed or unsigned and is thus ambiguous.

(2) Modelling: There is a problem with the parameter recovery of kappa (Suppl Fig 3). The authors conclude that it does not have explanatory value, but there could be other reasons for why the recovery is failing (i.e., a bug or no/insufficient variance in the data that kappa could load onto, in which case it is not a fair model to compare to). Even if the authors stick with Model 1, there is still a problem with beta which is not recovered well. What needs to be shown is that

- When data is simulated with kappa, kappa can be recovered, and the winning model is Model 3
- When data is simulated in the absence of kappa, the other two parameters can be recovered and the winning model is Model 1 (i.e. parameter recovery and model falsifiability)

If those points cannot be shown, then it cannot be concluded that Model 1 is necessarily a better fit to the data than Model 3. I also noticed that Model 1, despite having the lowest AIC, is the least likely in Table 2. AIC is well-known to favour simpler models. Would the conclusion hold for alternative model comparison approaches such as BIC or Bayesian model selection or cross-validation?

(3) I believe the fact that the cue and outcome phases were not separable should be mentioned earlier and deserves more attention. This affects the interpretation of all results. BOLD responses related to reward expectation, actual reward/no reward, as well as signed and absolute RPEs are all confounded at the same timepoint. In relation to this, it would be helpful if the specifics of how the various GLMs were set-up (signed/not signed) could be explained more clearly, possibly even in a supplementary figure.

(4) Valance should be valence

Reviewer #2 (Remarks to the Author):

Notes on "Dopamine release in human associative striatum in response to reward-prediction errors during reversal learning: Evidence from functional hybrid PET-MR" Grill et al. Nature Communications. 10-02-22 Resubmission.

The authors have adequately responded to the queries of this reviewer. Overall, the response is quite thorough with much helpful, new clarifying text and several follow-on analyses. This is now a much stronger paper.

Reviewer #3 (Remarks to the Author):

The authors did a thorough job addressing previous review questions in detail. The study results and its methodological approach are much better justified and well-reasoned, with several details listed in the extended supplementary material. While some questions remain about potential bias in ¹¹C-raclopride displacement, it is recognized that many pieces of evidence point to the validity of the described results and it is not practical to all be reconciled (unless an additional study is run). Overall, this is a carefully crafted study and manuscript including behavioral, fMRI and PET dynamic measures, that (combined) provide important insights to the scientific community.

A few additional points to address:

1) We suggest to clearly state assumptions that went into the analysis with the lp-ntPET model with respect to the onset time: Dopamine release is assumed to peak at the start of the volatile period because the model is set to describe this timeline and model the DA peak at this timepoint. The authors did not find differences between varying timelines, and so the exact time of the peak of the DA release may not be determined accurately (which is fine, but should be clear). Adding additional basis functions did not capture additional displacement, but if a set of basis functions was chosen that only started several frames later, would a similar result be obtained?

2) Why did the authors decide to temporally smooth the PET TACs before analysis? Did it improve analysis outcomes? Given the sparse timepoints on the PET (compared to fMRI), could this diminish outcome measures?

3) In all figures, please label colorbars. While they show numerical ranges, they are unlabelled but represent a variety of outcome variables.

4) The authors compare peak occupancy to RPE magnitude and sensitivity. Is it feasible to evaluate correlations between peak occupancy and fMRI responses in the spatially overlapping regions?

Response to reviews NCOMMS-22-36029A

We are pleased that all reviewers found that our first revision has substantially improved our manuscript "*Dopamine release in human associative striatum during reversal learning: Evidence from functional hybrid PET-MR*". Once again, we sincerely thank you for the thoughtful review of our work and your constructive suggestions for improvements. In this revised version, we have tried to address the additional points raised by reviewers 1 and 3. The corresponding changes in the main manuscript and supplement are highlighted in blue.

Reviewer #1

The authors have done a great job addressing most of my comments and I believe the manuscript has become clearer and stronger as a result. I have a few outstanding comments that have not been fully addressed.

(1) First, I am puzzled about why the authors prefer to stick to the terminology of a prediction error. Both, reinforcement learning theory by Sutton and Barto (and others) as well as the seminal findings by Schultz and colleagues in DA neurons (and many others since then), postulate that the prediction error is a signed response (i.e., positive response of DA neurons to outcomes that are more positive than expected and suppression of DA response to outcomes that are more negative than expected). I appreciate that other views are interesting and valid, but maybe using the same terminology of a RPE is then misleading because readers will assume the authors are talking about a signed response. Instead, here, the authors are focusing on an absolute signal which they call RPE amplitude. I suggest that the terminology is adapted so that it is intuitive for the reader and consistent with the literature. RPE magnitude is the same as surprise or absolute RPE which would be more appropriate. A magnitude could be signed or unsigned and is thus ambiguous.

We agree with the conceptual points made by the reviewer, and it is not our intention to mislead the reader. In the revised version, we have therefore edited the manuscript once more to better motivate and highlight the use of absolute RPE (absRPE). All instances where we previously used RPE is now changed to absRPE.

(2) Modelling: There is a problem with the parameter recovery of kappa (Suppl Fig 3). The authors conclude that it does not have explanatory value, but there could be other reasons for why the recovery is failing (i.e., a bug or no/insufficient variance in the data that kappa could load onto, in which case it is not a fair model to compare to). Even if the authors stick with Model 1, there is still a problem with beta which is not recovered well. What needs to be shown is that

- When data is simulated with kappa, kappa can be recovered, and the winning model is Model 3

- When data is simulated in the absence of kappa, the other two parameters can be recovered and the winning model is Model 1 (i.e. parameter recovery and model falsifiability) If those points cannot be shown, then it cannot be concluded that Model 1 is necessarily a better fit to the data than Model 3. I also noticed that Model 1, despite having the lowest AIC, is the least likely in Table 2. AIC is well-known to favour simpler models. Would the conclusion hold for alternative model comparison approaches such as BIC or Bayesian model selection or cross-validation?

We would like to thank the reviewer for this comment, and we fully understand the point. We have revisited all models, checked all code, and reran many analyses. We have also uploaded the model recovery scripts, models, and data to github (https://github.com/grillfilip/Dopamine-release-in-human-associative-striatum-during-reversal-learning/tree/main/model_recovery). In summary, we did not find a major bug in our models, but we made some smaller adjustments and additional analyses for this revision, which we detail in bullet points below. We think that it is likely that our experimental paradigm (a 50-minute task with a 30-minute stable condition) does not suit the complex models and that there is a failure to disentangle the parameters when using the mean and covariance of the fitted parameters (under the real data) to generate the synthetic data. As the reviewer mentions, there might not be enough variance in the data that the parameter can load onto. Therefore, we find it important to note that the conclusions we draw in this paper about human brain function do not change depending on which model we select. We do not make any mechanistic inferences regarding the models but only rely on the parameters (RPE sensitivity/learning rate) and generated quantities (absRPE) of Model 1 for correlations and as regressors for the fMRI. It has been discussed by Robert Wilson and Yael Niv that large discrepancies in e.g. learning rate from different models have minute changes on the fit of regressors to BOLD data (<https://journals.plos.org/ploscompbiol/article?id=10.1371/journal.pcbi.1004237>). This is also the case here. The correlations we perform on absRPE vs. dopamine occupancy as well as RPE sensitivity vs. dopamine occupancy hold no matter the choice of model (new Supplementary Fig. 1). We have now moved the model explanations from the methods section in the main manuscript to the supplementary materials.

Minor fixes and additional analyses:

-We realized that for Model 2, there was no explicit need to fit a separate beta for the volatile period. This model was therefore refitted with 3 parameters (α_1 , α_2 , and β).

-We identified that Model 3 (separate α for positive and negative outcomes) was likely underperforming due to the setting of an initial value of the expected value of choices that was incongruent with the other model assumptions (it was not set to zero). All models were checked and re-fitted with an initial value of 0.

-Because of updates to Models 2 and 3, all models were refitted to the data, compared against each other, and new model/parameter recovery analyses were conducted. This can be seen in

supplementary table 1 and 2. As previously, Model 2 can be confidently excluded from the model space.

-Model 3 now performs better (Supplementary Table 1). However, accounting for the number of parameters using BIC, Model 1 is still outperforming the other models. Note that BIC is more conservative than AIC as it punishes more strongly for the number of parameters. As such, AIC seldom finds evidence for the simplest model. The fact that more complex models do not outperform the simplest model in terms of BIC strongly suggests to us that their complexity is not justified by the data.

- Supplementary Fig.3a now shows the probability of choosing the index finger, the predicted probability under the real data (on policy), and the predicted probability under simulated data (off policy) for both Model 1 and Model 3. On policy prediction is nearly identical between the models. Off policy predictions show that Model 3 consistently underestimated the probability during the stable period (between trial 25 and 150) as well as underestimated the second reversal. This shows that the two models make very similar predictions and illustrates the fact that the extra parameters did not give Model 3 any substantial advantage when compared to the simpler Model 1 as shown by the BICs.

-Model recovery (Supplementary Fig. 3c) therefore now consists of Model 1, Model 3, Model 4, and Model 5. As before, the model recovery shows that only Model 1 recovers well. Having established that a bug was not the problem, we then set out to explore why this may be the case since it is indeed surprising that Model 3 (which performed well on the real data) did not recover. Looking at the distribution of learning rates for positive and negative outcomes for the simulated parameters (based on the mean and covariance of the real parameters) shows there is not much difference between them. This might be a reason why the simplest model with one learning rate better explains the data and is performing well on the data from Model 3. To test this idea further, we simulated two new datasets for Model 3, one where learning rate for positive outcomes was low and learning rate for negative outcomes was high, and one set where this was reversed (Supplementary Fig. 3d). Using a large learning rate for positive outcomes and a small learning rate for negative outcomes Model 3 was recoverable, confirming to us that, while the model is doing what it is supposed to do, the experimental paradigm (which was not optimal for modelling but optimized for the PET acquisition) does not fit the model. The recovery failure of Model 4 and 5 might be due to similar reasons. We investigated if the simulated data performed as expected, meaning that the learning rate should adapt trial to trial based on positive or negative outcomes. For positive outcomes the learning rate should increase, while for negative outcomes the learning rate should decrease. Supplementary Fig. 3e shows that the simulated data performs as expected (especially around the reversals), however we also notice that the magnitude with which the learning rate is modulated is quite low. The mean difference in modulation before and after reversals were 0.025 (Model 4) and 0.034 (Model 5). Again, these very small updates may be why the most simple model captures the data better. Parameter recovery is similar to the previous attempt; all parameters were reasonable recovered ($r > 0.7$) except κ for Model 4 and 5.

Overall, we think this second round of model comparison and control analyses greatly strengthens the manuscript even though our conclusions remain unchanged.

(3) I believe the fact that the cue and outcome phases were not separable should be mentioned earlier and deserves more attention. This affects the interpretation of all results. BOLD responses related to reward expectation, actual reward/no reward, as well as signed and absolute RPEs are all confounded at the same timepoint. In relation to this, it would be helpful if the specifics of how the various GLMs were set-up (signed/not signed) could be explained more clearly, possibly even in a supplementary figure.

We have revised the text to better foreshadow the task design and timing.

-Page 2, "Paradigm and Behavior", end of first paragraph: "The PET data was modeled to estimate DA release at the first reversal, i.e., the transition between stable and volatile phase. The fMRI analysis modeled individual events during the entire task. Cue, response and outcome response within each event were not separable."

-We now support the description of the GLMs with supplementary figure 7a and b.

-Page 9, "fMRI", end of last paragraph: "absRPE was orthogonalized with respect to the valence regressors, so that this analysis estimates voxels responding to absRPE, i.e. unexpectedness of a trial independent of the sign. The two GLMs are illustrated in supplementary Fig. 7a and b."

(4) Valance should be valence

Thank you, we have corrected this.

Reviewer #3

The authors did a thorough job addressing previous review questions in detail. The study results and its methodological approach are much better justified and well-reasoned, with several details listed in the extended supplementary material. While some questions remain about potential bias in ¹¹C-raclopride displacement, it is recognized that many pieces of evidence point to the validity of the described results and it is not practical to all be reconciled (unless an additional study is run). Overall, this is a carefully crafted study and manuscript including behavioral, fMRI and PET dynamic measures, that (combined) provide important insights to the scientific community.

A few additional points to address:

1) We suggest to clearly state assumptions that went into the analysis with the lp-ntPET model with respect to the onset time: Dopamine release is assumed to peak at the start of the volatile period because the model is set to describe this timeline and model the DA peak at this timepoint. The authors did not find differences between varying timelines, and so the exact time of the peak of the DA release may not be determined accurately (which is fine, but should be clear). Adding additional basis functions did not capture additional displacement, but if a set of basis functions was chosen that only started several frames later, would a similar result be obtained?

We thank the reviewer and agree with their suggestion to better state the a-priori assumptions that went into the PET analysis. In the revised manuscript, we have also summarized all included control analyses for the reader to underscore the point made by the reviewer that many pieces of evidence point to the validity of the described results. We do recognize that future research is needed to supplement our conclusions with different types of task designs.

We have made the following changes:

-Page 2: “The PET data was modeled to estimate DA release at the first reversal, i.e., the transition between stable and volatile phase, as dopamine release is assumed to peak at the start of the volatile period.”

-Page 8: “In the main results, 5 different gamma basis functions were fitted, hypothesis-driven to be consistent with the transition of the first reversal.”

-Page 8: “The following control analyses were performed: (1) the lp-ntPET model was fitted to TACs extracted from a priori defined striatal anatomical ROIs⁵⁴ to show that displacement was specific to associative striatum, mostly including caudate. (2) To investigate potential bias in the lp-ntPET model, a simulation analysis was performed (Supplementary Fig. 4a). (3) each of the 5 gamma functions was fitted at 4 different time points (2 frames immediately surrounding the reversal onset and 2 frames later into the first reversal, thus including 20 models; Supplementary Fig. 4b). (4) An extension of the lp-ntPET model was used to fit all reversal events simultaneously (Supplementary Fig. 5). (5) Potential confounding variables were investigated in relation to the occupancy estimation (Supplementary Fig. 6a). (6), single subject data for three representative subjects corresponding to low, medium, and high occupancy is shown in Supplementary Fig. 6b and c.”

In addition, we produced two more illustrations for the reviewer in response to this and the next comment that we do not feel are necessary to include in the manuscript. However, we can add them to the supplement if the reviewers and/or editors disagree with us.

a). Modelling a single set of basis functions a few frames later (at the third reversal) yields somewhat similar results as modelling the onset of the first reversal. However, assuming that

DA was released at the first onset, this analysis will be biased by Raclopride displacement that occurred before. We believe that the correct way to isolate a later onset is to do it with combined consecutive basis functions at each onset like in supplementary figure 5 which accounts for the displacement at each reversal. Please see inline figure 1 for a comparison of the voxelwise displacement tstat using a set of basis function at the first reversal (as in the main manuscript) and a set of basis functions at a later time point (the third reversal) as suggested in the comment.

Inline Fig1. Voxelwise tstat comparison of displacement at first reversal (top row) and third reversal (bottom row).

2) Why did the authors decide to temporally smooth the PET TACs before analysis? Did it improve analysis outcomes? Given the sparse timepoints on the PET (compared to fMRI), could this diminish outcome measures?

While there is a risk that temporal smoothing diminishes outcome measures because faster fluctuations are diminished, it minimizes the influence of single-frame extreme fluctuations due to e.g. residual scatter, residual randoms, and movements that are unlikely to reflect true displacement. In recent PET-fMRI studies that have focused on the temporal analysis of time-activity PET task data, others and we have used a similar approach (e.g. Hahn et al. 2021 Journal of Cerebral Blood Flow & Metabolism; Hahn et al. 2020 eLife; Stiernman et al. 2021 PNAS; Grill et al. 2021 Frontiers in Human Neuroscience).

For the reviewer, we have now re-analyzed the PET data without temporal smoothing, more reflective of a “typical” PET pre-processing pipeline. We find that using a small temporal smoothing filter increases the sensitivity of the displacement measure somewhat but does not affect the conclusions drawn from the data. Please see the inline figure 2 for a comparison of the voxelwise displacement tstat of smoothed versus unsmoothed data.

Inline Fig2. Voxelwise tstat comparison between smoothed (top row) and unsmoothed TACs (bottom row).

3) In all figures, please label colorbars. While they show numerical ranges, they are unlabelled but represent a variety of outcome variables.

The color bars have now been labeled with its corresponding statistics. Note that supplementary Fig. 7c does not have a statistics associated with it, these are group level maps contrasted against each other to visualize if any given voxel is more related to the covariate absRPE (previously RPE magnitude, see comment 1 from reviewer 1) or the contrast Reward>No Reward.

4) The authors compare peak occupancy to RPE magnitude and sensitivity. Is it feasible to evaluate correlations between peak occupancy and fMRI responses in the spatially overlapping regions?

This is feasible and an interesting idea. Thank you for the comment. In the revised supplementary, we have added linear and non-linear associations between absRPE fMRI beta estimates and DA occupancy from the significant PET ROI. Unfortunately, no significant association was observed. The same analysis was performed on the overlapping significant voxels as suggested by the comment and no significant association was observed either. Finally, we performed a voxelwise whole brain analysis using peak occupancy and its quadratic term as covariates to explain absRPE beta values. No significant voxels were observed. These analyses have been added to supplementary figure 8.

Page 1: “Whether the magnitude of DA release is proportional to fMRI signal magnitude is an open question because fMRI is an indirect measure not only reflective of neurotransmitter action but also many other signaling cascades.”

Page 4: “Correlations between individual differences in BOLD signal and DA occupancy were not significant (Supplementary Fig. 8).”

Page 5: “What remains unclear from this human PET modeling work is whether and how DA PET signals and the BOLD signal are mechanistically linked, both where they overlap and where

they do not overlap. Here, we explored the possibility of an association between BOLD and DA occupancy but found no significant correlations (Supplementary Fig. 8). A dose response relationship between raclopride and cerebral blood volume has previously been observed in non-human primates, providing ample evidence for neurovascular coupling of D2/D3 receptors⁴⁰. However, using a ligand specific to D2/D3 receptors in humans with a cognitive task instead of selective pharmacological manipulations, we cannot conclusively speak to the influence of DA release on the BOLD signal since this is dependent on what type of receptor (D1-like or D2-like) DA binds to; in the current study it is likely that DA binds to a mix of receptor types. Moreover, also non-DAergic signals are captured in the BOLD signal. Thus, BOLD activations that are not accompanied by DA release might indicate additional neurotransmitter processes within a cortico-striatal network. Ideally, future hybrid PET-MR studies would implement multi-tracer studies or combine cognitive with pharmacological challenges in order to capitalize on the unique possibility of hybrid PET-fMRI to understand network-wide brain dynamic in terms of underlying neurotransmitter action.”

Reviewer #1 (Remarks to the Author):

The authors have done a great job addressing my remaining concerns and I have no further comments.

Reviewer #3 (Remarks to the Author):

The authors have done a nice job addressing previous comments and their detailed work has made this a very nice manuscript. I have no additional comments to add.